# EXPLAINABLE RECOMMENDER WITH GEOMETRIC INFORMATION BOTTLENECK

## ABSTRACT

Explainable recommender systems have attracted much interest in recent years as they can explain their recommendation decisions, enhancing user trust in the systems. Most explainable recommender systems rely on human-generated rationales or annotated aspect features from user reviews to train models for rational generation or extraction. The rationales produced are often confined to a single review. To avoid the expensive human annotation process and to generate explanations beyond individual reviews, we propose an explainable recommender system trained on reviews by developing a transferable **G**eometric **I**nform**A**tio**N** bo**T**tleneck (**GIANT**), which leverages the prior knowledge acquired through clustering on a user-item graph built on user-item rating interactions, since graph nodes in the same cluster tend to share common characteristics or preferences. We then feed user reviews and item reviews into a variational network to learn latent topic distributions which are regularised by the distributions of user/item estimated based on their distances to various cluster centroids of the user-item graph. By iteratively refining the instance-level review latent topics with **GIANT**, our method learns a robust latent space from text for rating prediction and explanation generation. Experimental results on three e-commerce datasets show that our model significantly improves the interpretability of a variational recommender using the Wasserstein distance while achieving performance comparable to existing content-based recommender systems in terms of rating prediction accuracy.

## 1 INTRODUCTION

Typically, a recommender system compares users' preferences with item characteristics (e.g., item descriptions or item-associated reviews) or studies user-item historical interactions (e.g., ratings, purchases or clicking behaviours) in order to identify items that are likely of interest to users. In addition to predictive performance, interpretable recommenders aim to give rationale behind the rating given by a user on an item (Ghazimatin et al., 2020; Zhang et al., 2020). Most existing interpretable recommenders can either generate rationale or extract text spans from a given user-item review as explanations of model decisions. Both rationale generation and extraction require annotated data for training, e.g., short comments provided by users explaining their behaviours of interacting with items, or annotated sentiment-bearings aspect spans in reviews (Zhang et al., 2014; Ni et al., 2019; Chen et al., 2019; Li et al., 2020; Tan et al., 2021a).

We argue that generating explanations based on a specific user-item review document suffers from the following limitations. First, some reviews may be too general to explain the rating, rendering them useless for explanation generation. For example, the review '*I really like the smartphone, will recommend it to my friends*' does not provide any clue why the user likes the smartphone. Second, features directly extracted from a review document may fail to reflect some global properties which can only be identified from implicit user-item interactions. For example, meaningful insights could still be derived from reviews towards items that are not directly purchased/rated by a user but preferred by other like-minded users. Finally, explanation generation model from user/item reviews are often supervised by human-annotated rationales, which are labour-intensive to obtain in practice.

To address the aforementioned limitations, we propose an AutoEncoder (AE) framework with variational **G**eometric **I**nform**A**tio**N** bo**T**tleneck (**GIANT**) to incorporate the prior from user-item interaction graph to refine the induced latent factors of user and item, and generate explanations in an unsupervised manner. For a user-item pair, all reviews written by the user and reviews posted on the

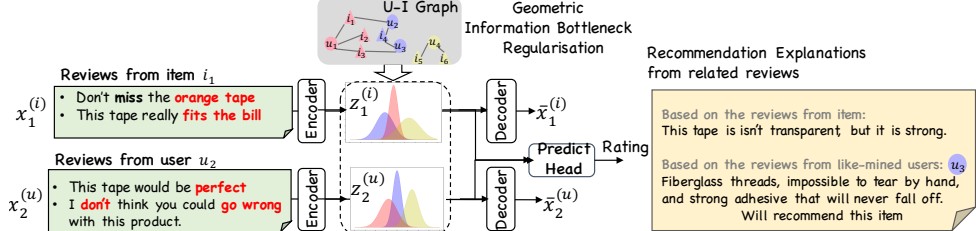

Figure 1: An encoder-decoder structure with a geometric information bottleneck regularisation, which is derived from the U-I interaction graph and used as a prior imposed on $Z$. The latent variable $z$ thus can capture the geometric affiliations in graph and be the soft cluster assignment distributions. It enables the use of like-minded users $u_3$ and similar items $i_2$ and for generating explanations beyond the input user-item pair $(i_1, u_2)$. Existing rationale extraction methods can only extract indicative words (shown in red) from the given review pair. Our method can generate rationale from reviews written by other like-minded users or on similar items, which are assigned into the same latent dimension/cluster.

item are fed into two separate encoders to infer latent factors, and predict rating based on the match of their latent factors. The latent variables are supposed to capture the key semantic information to recover the original review text written by the user on the item. Different reviews are assigned to different latent dimensions according to posterior distributions and we extract the representative words in each dimension to summarize the cluster topic. The explanations are thus from the reviews related to the topic of the assigned cluster (See in §6.3).

Our proposed framework is illustrated in Figure 1. The geometric regularisation refers to the cluster-based distance with Gaussian variance, which is derived by firstly clustering the users/items in the user-item interaction graph, and then calculating the distribution of a user/item as its distance to each cluster centroid by a Gaussian kernel (§ 4.1). To impose the regularisation on the latent variables of the AE framework taking the set of user reviews and item reviews as input, we apply the KL divergence to minimize the discrepancy between the induced posterior distribution of users $z^{(u)}$, and items $z^{(i)}$, and the geometric regularisation as the prior (§4.2), after linking the cluster to the review text encoder via prior-centralisation (§4.2).

Experimental results on the three commonly used benchmarking datasets show that the proposed method achieves performance comparable with several strong baselines in recommendation. Moreover, the quantity and the quality analysis in interpretability show that our method can generate coherent, diverse and faithful explanations.

## 2 RELATED WORK

We review recommender systems, with particular attention to those built on Variational Autoencoder (VAE) (Kingma & Welling, 2014) or offering explanations of recommendation decisions.

**Recommender with variational autoencoder** Text data such as user reviews, item or brand descriptions could be important for developing a high-quality recommender system as they can be exploited to address the sparsity issue in user-item interactions. HFT (McAuley & Leskovec, 2013) and CTR (Wang & Blei, 2011) adopted Latent Dirichlet Allocation (LDA) (Blei et al., 2003) to extract latent topics from review text. VAE can also be used for text modeling due to its ability in extracting latent and interpretable features (Fei et al., 2021; Truong et al., 2021; Wang et al., 2020). Truong et al. (2021) argued that the commonly used isotropic Gaussian in VAE is over-simplified and proposed BiVAE by introducing constrained adaptive prior (CAP) for learning user- and item-dependent prior distributions. More recently, review information is used to disentangle the latent user intents at the finer granularity, for example, DisenGCN (Ma et al., 2019a) and DisenHan (Wang et al., 2020) used the graph attention mechanism to differentiate multiple relations and features.

**Explainable Recommender System** One popular interpretable method is to train a language generation model with the ground-truth explanations supplied, which can be the first sentence of a given review or human-annotated text spans in the review text (Li et al., 2017; Chen et al., 2019; Ni et al.,

2019; Li et al., 2022). Feature-level explanations aim to provide the important item features the users care about when making decisions (Zhang & Chen, 2020). Explicit Factor Model adopted Matrix Factorization (MF) to predict review ratings and aligned each dimension of latent factors of the MF results with some item features, thus explaining the recommendation decision by the corresponding features (Zhang et al., 2014). Other approaches firstly processed the review text to extract user-aspect-opinion and item-aspect-opinion tuples. They then modeled the interactions between the two tuples to derive the rating and the indicative features (Chen et al., 2016; Wang et al., 2018; Tan et al., 2021b; Chen et al., 2020). However, the aforementioned approaches suffer from two major issues. Some approaches require aspects to be extracted first in order to provide aspect-related predictive features as interpretations, thus heavily dependent on the quality of the aspect extractors employed; while others rely on human-annotated rationales to train supervised language generators for explanation generation. Our proposed approach can instead generate explanations with the only supervision signals coming from review ratings.

## 3 GEOMETRIC INFORMATION BOTTLENECK

To consider implicit user-item interactions for developing an interpretable recommender system, we propose to incorporate the geometric regularisation derived from user-item interaction graphs to learn the latent factors of users and items from review text in a variational network. Although various work has been proposed to adopt richer priors to infer a more complex and realistic posterior in standard VAEs (Tomczak & Welling, 2017; Zhao et al., 2017), our solution is different from existing approaches as we need to impose the prior knowledge learned from the graph modality to constrain the learning of latent variables in the text modality. In what follows, we show how this can be done under the theory of information bottleneck.

Based on the Information Bottleneck (*IB*) theory (Tishby & Zaslavsky, 2015), we can train an encoder-decoder architecture $x \xrightarrow{encode} z \xrightarrow{decode} \bar{x}$, where $x \in X$, $z \in Z$, $\bar{x} \in \bar{X}$ are the input, the hidden, and the reconstructed representations, respectively, in order to preserve the meaningful information about $\bar{X}$ in $Z$ while maximally compress the information of $X$. We achieve it by maximising the following equation:

$$O_{IB} = I(\bar{X}; Z) - \beta \cdot I(X; Z), \tag{1}$$

where $I(\cdot)$ denotes the mutual information, $\beta$ is a Lagrange multiplier. The second term in Eq. (1) is to maximally compress $X$, equivalent to minimize the mutual information between $X$ and $Z$, denoted as $I(X; Z) \leq I_c$, $I_c$ is the upper bound.

In this paper, the goal is to build a mechanism which is able to compress the information across different modalities, i.e., the text and graph, while keep the mutual information between the input $X$ and compressed representation $Z$ bounded by an upper bound as well.

Intuitively, we assume the $n$-th input $x_n$ has representations $x_n^t \in X^t$ and $x_n^g \in X^g$ under different modalities[1] satisfying the constraint, $I(X^t; X^g) \geq I_x$, since $X^t$ and $X^g$ are derived from the same input, they should be relevant. Then, two independent well-trained encoder-decoder architectures in the two modalities optimised by Eq. (1) are constrained by $I(X^t; Z^t) \leq I_c$ and $I(X^g; Z^g) \leq I_c$, respectively. We derive a new upper bound for the multi-modality IB as follows:[2]

$$I(X^t; Z^g) \leq H(X^t) - H(X^g) + H(X^g|X^t) + I_c \tag{2}$$

where $H(\cdot)$ and $H(\cdot|\cdot)$ are entropy and conditional entropy, respectively. Therefore, we can maximise the objective function using the input from the text modality, with a regularisation from another modality such as a user-item interaction graph, but still maintain a new upper bound. [3]

$$O_{IB}^t = I(\bar{X}^t; Z^t) - \beta \cdot I(X^t; Z^g). \tag{3}$$

According to (Alemi et al., 2017), the above lower bound can be approximated by applying the reparameterisation trick (Kingma & Welling, 2013) with a random Gaussian noise $\epsilon$:

$$O_{IB}^t = \Sigma_{n=1}^N \mathbb{E}_{\epsilon \sim p(\epsilon)}[-\underbrace{\log q(\bar{x}_n^t|f(x_n^t, \epsilon))}_{\text{Recons. term: } Q(x_n^t)}]/N + \beta \cdot \underbrace{\text{KL}(p(Z^t|x_n^t, \epsilon)|r(Z^g))}_{\text{KL}-\text{div. term: } \text{KL}(Z^t|Z^g)}, \tag{4}$$

---

[1] In the rest of paper, we use $t$ for text and $g$ for graph.

[2] The proof is shown in the Appendix B.

[3] The proof is shown in the Appendix B.

where $x_n^t$ is the training sample from one modality $X^t$ (e.g., text), $\bar{x}_n^t$ is the VAE-based reconstruction output, $q(\cdot)$ and $p(\cdot)$ are the posterior probabilities estimated by decoder and the encoder, respectively, $r(Z^g)$ is the estimated distribution from another modality (e.g., user-item interaction graph). Here, Eq. (4) has a similar form as $\beta$-VAE which contains a reconstruction term $Q(x_n^t)$ defined by the posterior probability and a KL-divergence term $\mathrm{KL}(Z^t|Z^g)$ regularised by the prior from another modality $Z^g$. By this equation, we are able to guarantee that the information bottleneck between two modalities are:

- **Transferable** − According to Eq. (2), the information bottleneck between modalities is still constrained by the global information bottleneck, i.e., we can take the bound from one modality to guide the training on the other.
- **Practicable** − One challenge in the training of variational models is the choice of the prior distribution. By Eq. (2), we do not have to utilise a pre-defined prior but instead use the posterior constraints from other modalities, which are more flexible and efficient.
- **Interpretable** − The alignment between representations from different modalities and their corresponding probabilities defined in the geometric space makes it possible to use modalities offering better interpretability, such as review text, to explain the black-box features from other modalities.

## 4 GIANT FRAMEWORK

We propose a novel explainable recommender built on VAE with the geometric information bottleneck. For a given $(user, item)$ pair, the input to our recommender is a set of reviews written by the user and a set of reviews on the item. As each user or item is associated with multiple reviews, and some reviews might be very long, the Convolutional Neural Network (CNN) is used to encode reviews due to its efficiency in encoding long sequences, as have been previously studied in Chen et al. (2018); Zheng et al. (2017). We then generate the user or item representation by aggregating their associated multiple reviews by the attention mechanism. The final contextual vector for user $u$ and item $i$ is denoted as $x^{(u)}$ and $x^{(i)}$, respectively.

Given the contextual vectors, we employ two geometric regularised variational networks to infer their latent factors, in order to better capture their latent semantic topics. According to Eq. (2), we use graph-based $Z^g$ as the transferable information bottleneck to regularise the optimisation of the text-based $Z^t$. To ensure feasibility and robustness, we propose an initialisation trick called prior-centralisation which uses the Gaussian kernel based geometric estimation to regularise the initialisation and optimisation of the encoders based on the transferable information bottleneck. In the experiments section, we verify the interpretability of our proposed method by the alignment between the two latent feature spaces from graph and text modalities.

### 4.1 DERIVING PRIORS FROM USER/ITEM CLUSTERS IN INTERACTION GRAPH

In the user-item bipartite graph $\mathcal{G}$, a node can be either a user or an item. For each user-item pair, an edge is created if the user has previously rated on the item and the rating score is higher than the average rating score calculated across all user-item ratings.[4] We use the *LightGCN* (He et al., 2020) to encode our interaction graph and obtain the user and item embeddings.

Once user/item embeddings are learnt, we apply K-means on the learned node embeddings to derive the cluster centroid vectors, denoted as $\boldsymbol{C}_k^{(u)}$ for users and $\boldsymbol{C}_k^{(i)}$ for items, $k \in \{1, 2, \cdots, K\}$. We use the Radial Basis Function (RBF) kernel to compute the distance between a user or item with the cluster $k$ as the probability of assigning the user or item $j$ to the $k$-th cluster, $\rho_{jk}$:

$$\rho_{jk}^g = \frac{\exp\left(-\left\|\boldsymbol{e}_j^g - \boldsymbol{C}_k\right\|^2 / 2\alpha^2\right)}{\sum_{k'=1}^K \exp\left(-\left\|\boldsymbol{e}_j^g - \boldsymbol{C}_{k'}\right\|^2 / 2\alpha^2\right)}, \tag{5}$$

where $\boldsymbol{e}_j^g$ is graph node embedding for user or item $j$, $\boldsymbol{C}_k^g$ is $k$-th cluster centroid vector for user embeddings or item embeddings, $\alpha$ is a hyper-parameter adjusted according to the data density.

---

[4] We have also experimented with the creation of edges for each observed user-item pair, but observed worse performance.

The cluster-based distribution $\rho^g = \{\rho^g_{jk}\}$ is then used to regularise the latent factors for users and items, as will be shown in § 4.2. Moreover, we have $\Sigma_k \rho^g_{jk} = 1$, that is, the distribution of $\{\sqrt{\rho^g_{jk}}\}$ resides in a hyper spherical cap area on an uniform $K$-ball, weakly approximating to a Gaussian distribution when $K$ is large (Diaconis & Freedman, 1987). We therefore obtain a latent vector $Z^g$ and a corresponding geometric space where the difference on each basis follows a Gaussian prior.

## 4.2 A POSTERIOR OVER CONTROLLABLE DISTRIBUTIONS FROM TEXT

In section 4.1, we estimate the distribution of $Z^g$ from a user-item bipartite graph. The next step is to optimise the objective of Eq. 4 to infer the latent factors of users and items, $z^{(u)}$ and $z^{(i)}$ in text modality. In this section, we perform the following two steps:(1) minimising the lower bound of $O^t_{IB}$ by a RBF-kernel estimated posterior probability; and (2) a prior-centralisation term which encourages the encoder weight matrix in the variational network to be closer to the cluster centroid representations learned in the interaction graph, so that the learnt latent variables naturally capture the cluster-based distances.

**Optimising Conditional Probability by Kernel Density** For the $Q(x^t_n)$ in Eq. (4) which is defined as the conditional probability of $\log q(\bar{x}^t_n | f(x^t_n, \epsilon))$, we use the Nadaraya-Watson estimator (Hall et al., 1999):

$$Q(x^t_n) = \log \frac{\hat{p}(\bar{x}^t_{n,\epsilon}, x^t_n)}{\hat{p}(x^t_n)} = \log \frac{\frac{1}{N}\sum_{j=1}^N \kappa(\frac{x^t_n - x^t_j}{h}) \cdot \kappa(\frac{\bar{x}^t_{n,\epsilon} - x^t_j}{h})}{\frac{1}{N}\sum_{j=1}^N \kappa(\frac{x^t_n - x^t_j}{h})}, \tag{6}$$

where $\kappa(\cdot)$ is a kernel function for density estimation, $\bar{x}^t_{n,\epsilon}$ is the reconstructed $x^t_n$ with random noise $\epsilon$. In practice, we choose the RBF kernel $\kappa = \exp(-||x - x'||^2))$ and ignore the denominator since the goal is to optimise the decoding of $\bar{x}^t_n$, then we are able to obtain the following bound by the triangle inequality in the learned metric space as $Q(x^t_n) \propto \log(\exp(-||\bar{x}^t_{n,\epsilon} - x^t_n||^2))$. To simplify the computation, we choose natural logarithm function in optimisation and obtain a mean squared error based objective function $O^t_{IB}$[5]:

$$O^t_{IB} = \frac{1}{N}\Sigma_{n=1}^N \mathbb{E}_{\epsilon \sim p(\epsilon)}[||\bar{x}^t_{n,\epsilon} - x^t_n||^2] + \beta \cdot \text{KL}(p(Z^t|x^t_n, \epsilon)|r(Z^g)), \tag{7}$$

where $x^t_n \in \{x^{(u)}\} \bigcup \{x^{(i)}\}$ denote the input user and item representations, and $\bar{x}^t_{n,\epsilon}$ denote the reconstruction. By minimising the loss, we can preserve the key local similarity of nearby representations while ensuring different representations from different feature characteristics (Czolbe et al., 2020; Wang et al., 2004). Note that we stop the gradient back-propagation of the input $x^t_n$ and optimise the encoder-decoder parameters only, because the updating of input representations might reduce the relevance between modalities, break the assumption of $I(X^t, X^g) \geq I_x$, and lead to two independent VAEs as a degeneracy of transferable information bottleneck given by Eq. 2.

**Minimising Distribution Discrepancy** To minimise the second term of KL-divergence in $O^t_{IB}$, we need to project $Z^g$ and $Z^t$ to a metric space with the same size. Inspired by (Van der Maaten & Hinton, 2008), a Student $t$-distribution is used to mitigate the crowding problem. Here, we use the softmax activation with temperature $\tau$ to adjust the distribution tail and map its values to the range of $[0, 1]$. The $k$-th dimension of a latent variable $z^t_{nk}$[6] denotes the probability of the corresponding user or item being assigned to the $k$-th cluster:

$$\eta^t_{nk} = \frac{1 + e^{\varphi(z^t_{nk})/\tau}}{\sum_{k'=1}^K e^{\varphi(z^t_{nk'})/\tau}}, \quad \text{KL}(p(Z^t|x^t_n, \epsilon)|r(Z^g)) = \sum_n \sum_{k=1}^K \eta^t_{nk} \log \frac{\eta^t_{nk}}{\rho^g_{nk}}, \tag{8}$$

where $\varphi(z^t_n)$ is a linear transformation of $z^t_n$. Here we minimise the KL-divergence between $\rho^g_{jk}$ (in the user-item interaction graph space) and $\eta^t_{jk}$. Therefore, we rewrite the $O^t_{IB}$ as:

$$O^t_{IB} = \frac{1}{N}\Sigma_{n=1}^N \mathbb{E}_{\epsilon \sim p(\epsilon)}[||\bar{x}^t_{n,\epsilon} - x^t_n||^2] + \beta \cdot \sum_n \sum_{k=1}^K \eta^t_{nk} \log \frac{\eta^t_{nk}}{\rho^g_{nk}}, \tag{9}$$

---

[5]The detailed discussion can be found in Appendix C.

[6]Here, $n$ can either be a user index $u$ or or an item index $i$.

**Prior-centralisation: Linking Clusters with Encoder Weights**   In the encoding progress, we obtain the latent variable $z_n^t = f(x_n^t, \epsilon)$ by applying the reparameterisation trick (Kingma & Welling, 2013). However, due to the randomness in initialisation, the learned distribution of $z_n^t$ might be chaotic. Therefore, in our one-layer MLP based encoder, we propose a prior-centralisation trick which builds the connection between the pre-trained clusters and the encoder by making the encoder weights close to the cluster centroids $\boldsymbol{C}^g \in \mathbb{R}^d$ obtained from the graph:

$$\mathcal{R}_{centroid} = \sum_{k=1}^{K} \left\| W_{en}^k - \boldsymbol{C}_k^g \right\|^2 \tag{10}$$

where $W_{en}^k$ denotes the $k$-th column of the encoder weight matrix $W_{en}$ [7] and $\boldsymbol{C}_k^g$ denotes the representation of the $k$-th cluster centroid vector. Note that the $R_{centroid}$ will force the distribution of $z_n^t$ approximate to the prior and ignore the input. Therefore, in practice, this regularisation is only deployed in the initial training epochs.

**Rating Prediction and Final Objective Function**   After obtaining the regularised latent contextual representations $\boldsymbol{z}^{(u)}$ and $\boldsymbol{z}^{(i)}$, we add their corresponding ID features to obtain the final user and item embeddings $\zeta_u$ and $\zeta_i$ for rating prediction, $\zeta_u = W_u \boldsymbol{z}^{(u)} + \epsilon_u, \zeta_i = W_i \boldsymbol{z}^{(u)} + \epsilon_i$, where the ID features $\epsilon_u$ and $\epsilon_i$ are generated by feeding the user ID and item ID to an embedding layer. Inspired by the latent factor model in recommendation systems, we introduce the global biases for users and items in the final prediction layer as $\hat{r}_{ui} = f_{\text{cls}}(\zeta_u, \zeta_i) + b_u + b_i$, where $f_{\text{cls}}$ combines the user and item features into a scalar, $b_u$ and $b_i$ are bias derived from $\epsilon_u$ and $\epsilon_i$. The regression loss $\mathcal{L}_r$ of the predicted rating is calculated as the MSE on the given user-item pair. Combining all the components above, we derive the training objective as follows:

$$\mathcal{L} = \mathcal{L}_r + O_{IB}^t + \mathcal{R}_{centroid} \tag{11}$$

## 5 EXPERIMENTS

### 5.1 EXPERIMENTAL SETUP

**Datasets and Metrics**   The evaluation datasets include *BeerAdvocate* (McAuley et al., 2012) and two amazon review datasets, *Digital Music* and *Office Products* (He & McAuley, 2016) [8]. We use Root Mean Squared Error (RMSE) and Mean Absolute Error (MAE) to evaluate the rating prediction accuracy. [9]

**Baselines**   We compare with several open-source recommenders, including *HFT* (McAuley & Leskovec, 2013), *DeepCoNN* (Zheng et al., 2017), *NARRE* (Chen et al., 2018). *HFT* combines reviews with ratings and uses an exponential transformation function to link review text and the ratings. *DeepCoNN* uses a shared layer for interaction modeling the users and items, which is on top of the two encoders for the users and items, respectively. *NARRE* introduces review-level attentions to select important reviews and incorporates the user and item IDs as discriminative features in rating prediction. Besides, we apply an encoder-decoder *AutoEncoder* on top of *NARRE* to learn the latent variable as a baseline to highlight the difference in our proposed regularisation.

**Training Procedure**   We train the model parameters by minimising the objective function defined in Eq. (11). The $\beta$ for the KL divergence term is set to be 0.01 and we introduce the $L_2$ regularisation for all the model parameters and the weight is 0.001. [10]. We only train the prior-centralisation term for 0.5 proportion to approximate the cluster centroids for the later training. The KL term in $O_{IB}$ should be introduced later until the encoder centroid is well trained. To do so, we follow a similar cyclical schedule (Fu et al., 2019) to gradually adjust the anneal factor $\lambda$ in each epoch. Specifically, we first train the model without the KL term for 0.5 proportion, then anneal it from 0.5 to 1 for 0.25 proportion, and finally fix $\lambda = 1$.

---

[7]We also experiment with minimising the distance between decoder weight and the cluster centroid, as the encoder and decoder are asymmetrical.

[8]The dataset statistics and pre-processing details can be found in Appendix A.1.

[9]The ranking-based evaluation results, i.e., the overlapping between the recommended items and the target items are shown in Appendix D.1.

[10]The hyper-parameter settings are described in Appendix A.2.

| Models | BeerAdvocate | | Digital Music | | Office Products | |
|---|---|---|---|---|---|---|
| | RMSE($\downarrow$) | MAE($\downarrow$) | RMSE($\downarrow$) | MAE($\downarrow$) | RMSE($\downarrow$) | MAE($\downarrow$) |
| *HFT* | 79.81 | 62.42 | 96.42 | 74.76 | 89.46 | 68.57 |
| *DeepCoNN* | 77.28 | 59.45 | 94.69 | 71.03 | 85.14 | 64.82 |
| *NARRE* | 76.80 | 58.94 | 93.69 | 69.30 | 84.40 | 63.40 |
| *AutoEncoder* | 75.94 | 58.93 | 93.97 | 69.13 | 85.03 | 64.32 |
| *GIANT* | 75.36* | 57.87* | 92.87* | 68.68* | 84.32 | 62.05* |

Table 1:Performance comparison in RMSE (%) and NAE (%) for all methods. * denotes the statistical significance for $p < 0.01$, compared to the best CNN-encoder based baseline.

## 5.2 RATING PREDICTION RESULTS

### 5.2.1 MAIN RESULTS BY COMPARING TO BASELINES.

The rating prediction results of our model in comparison with baselines on all datasets are given in Table 1. We have the following observations. (1) *DeepCoNN* built on the stack of non-linear neural networks for review semantic modeling outperforms *HFT* which leverages an exponential transformation function to link topic distributions in review text and latent factors derived from ratings. This shows the superiority of deep learning for feature extraction. (2) *AutoEncoder* gives slightly better performance than *NARRE* in *BeerAdvocate*, which shows its effectiveness of extracting key contextual information for rating prediction. (3) Our method consistently outperforms all the baselines and the improvement is more predominant on the *BeerAdvocate*, which has the smallest sparsity. The results demonstrate the effectiveness of our proposed information bottleneck regularisation applied on the latent semantic space. By doing this, the users and items can be grouped into different clusters according to the interaction data, which is not the case in other textual CNN-based recommenders.

### 5.2.2 PERFORMANCE CONTRIBUTIONS FROM VARIOUS MODULES.

**a) Effects of different loss terms** We study the effects of the three loss terms in rating prediction accuracy and *Diversity* of the latent variables. As variational networks could easily collapse into an unconditional generative model, i.e., in the extreme case, all the input will be mapped into the same latent code (Ma et al., 2019b). We use the dimension index whose corresponding latent value is the maximum as the cluster assignment ID. We then derive the cluster assignment results $\mathcal{A} \in \mathbb{R}^{N \times K}$, where $N$ is the number of test samples, $K$ is the cluster number. The diversity is calculated based on entropy $H(X) = -\sum_{k \in K} p(k) \log(p(k))$, where $p(k)$ is the fraction of the number of samples falling into the $k$-th cluster among all the samples. A larger value means a better diversity. We record the largest diversity between user and item latent variables. The results are shown in Table 2.

| Variants | BeerAdvocate | | Digital Music | | Office Products | |
|---|---|---|---|---|---|---|
| | RMSE($\downarrow$) | Div($\uparrow$) | RMSE($\downarrow$) | Div($\uparrow$) | RMSE($\downarrow$) | Div($\uparrow$) |
| **Full Model** | 75.36 | 1.89 | 92.87 | 3.46 | 84.32 | 1.58 |
| **-w/o $\mathcal{R}_{centroid}$** | 75.96 | 0.04 | 92.62 | 0.07 | 84.58 | 0.02 |
| **-w/o KL term** | 75.64 | 1.43 | 93.83 | 2.86 | 84.23 | 1.21 |
| **-w/o $Q(x_n)$** | 76.02 | 1.04 | 93.95 | 3.10 | 84.94 | 0.93 |
| **$Q(x_n)$ with Cosine** | 75.35 | 0.11 | 92.85 | 1.76 | 84.97 | 0.07 |

Table 2: *RMSE*(%) and *Diversity* among model variants.

We observe that the removal of $Q(x_n^t)$ leads to the largest performance degradation. After removing $\mathcal{R}_{centroid}$, the latent variable diversity shrinks to near zero, indicating nearly all the latent variables fall into the same cluster. This highlights the capability of our prior-centralisation term in enabling the latent variables $z_n^t$ effectively reflect the soft cluster assignments. We also found that replacing the MSE in $Q(x_n^t)$ with the cosine similarity ($1 - cosine$) reduces the latent variable diversity significantly. This can be partly explained by the fact that the cosine similarity focuses on measuring the angle between the input $x_n^t$ and the reconstructed output $\bar{x}_n^t$, ignoring the magnitude of the vectors which is however important in our case.

**b). Effect of our proposed information bottleneck regularisation** We compare our approach with *StandPrior*, *WassersteinVAE* and *IndivPrior*: *StandPrior* (aka. *StandVAE*) uses two VAEs for users and items respectively, each with the standard Gaussian distribution $\mathcal{N}(0, 1)$ as the prior. *WassersteinVAE* differs from the *StandPrior* in using the Wasserstein metric (Tolstikhin et al., 2017) to calculate the distribution discrepancy between posterior $\eta$ and prior $\rho$. *IndivPrior* assigns a separate Gaussian prior to each user or item with its mean value calculated from the user's or item's corre-

sponding representation derived from the user-item interaction graph, i.e., $p(z^{(u)}) = \mathcal{N}(W_u \xi u, \boldsymbol{I})$, $p(z^{(i)}) = \mathcal{N}(W_i \xi i, \boldsymbol{I})$, where $\xi u \in \mathbb{R}^d$, $\xi i \in \mathbb{R}^d$ denote the user and item node representations in GCN learning, $W_u$ and $W_i$ are learnable parameters in a linear layer of size $(d \times d)$, and $d$ is the dimension of graph node features.

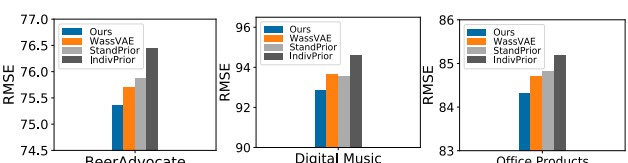

Figure 2: *RMSE*(%) with different information bottleneck regularisation. Our method achieves the best, followed by *StandPrior* and *WassersteinVAE*. *IndivPrior* is the worst, even worse than *AutoEncoder*.

The rating prediction results are shown in Figure 2. *WassersteinVAE* and *StandPrior* demonstrate overall better results than *IndivPrior* and all the baselines in Table 1. The improvement can be explained by the nature of VAE, which is particularly beneficial when dealing with sparse data where few observations are available. However, *IndivPrior*, which imposes a separate Gaussian prior on individual user/item, shows worse performance compared to the other three variational frameworks, and is even worse than *AutoEncoder*. Our model benefits from a rich prior separated from different user/item clusters, while avoiding using a global normal prior (*StandPrior* and *WassersteinVAE*) or a separate instance-level prior for each user/item (*IndivPrior*).

## 6 LATENT VARIABLE INTERPRETABILITY EVALUATION

### 6.1 CLUSTER SEPARABILITY AND COHERENCE

As each dimension of the latent variables $z_n^t$ corresponds to a cluster, for the $k$-th dimension, we can search for its relevant reviews which have the highest value in $z_{nk}^t$ and list the most frequent words in the review set as the representative topic words [11]. Representative words in randomly selected three clusters on *BeerAdvocate Dataset* are shown in Table 3. For our model, the most prominent words are different across different clusters (thus being colored), i.e., '*brown roasted*', '*pine, caramel*' and '*good flavour*'. We find it hard to see clear topic separations from the *WassersteinVAE* results. For example, Cluster 2 and 3 share the same 3 words, '*sweet*', '*light*' and '*malt*' out of their top 4 words. In addition, '*carbonation*' in Cluster 2 and '*whitehead*' in Cluster 3 are both relating to beer foam. The results show that with the incorporation of the priors derived from user/item clusters, our proposed approach is able to learn latent variables in the review semantic space which can produce better separable topic clusters.

| GIANT | WassersteinVAE |
|---|---|
| **pour**, color, **brown**, feel, **roasted**, malts, almost, moderate, coffee, dark | light, color, malt, **glass**, poured, **drink**, sour, pour, mouthfeel, **alcohol** |
| **pine**, **caramel**, lacing, **citrus**, mouth-feel, hint, ipa, note, body, strong | sweet, **coffee**, light, malt, aroma, **car-bonation**, pour, mouthfeel, hint, color |
| **good**, **flavour**, dark, **better**, brew, style, bad, Canadian, nose, drinking | aroma, light, sweet, malt, **white head**, mouthfeel, **nose**, hint, **brew**, note |

Table 3: The most prominent words (sorted by occurrence frequency) in three randomly selected clusters from **GIANT** and *WassersteinVAE*. We highlight the top 3 words that are not found in the other two clusters.

### 6.2 COMPREHENSIVENESS EVALUATION BY PERTURBING ON LATENT VARIABLES

To explore the importance of our identified clusters, i.e., the latent dimension with larger value, we calculate the performance change before and after removing the specific dimension and define *Comprehensiveness* as: $\frac{\sum_i^N \left( r(z_i^t) - r(z_i^t / z_i^{t[k]}) \right)^2}{N}$, Where $r(\cdot)$ is the predicted rating, $z_i^t$ is the latent variable for $i$-th evaluated user-item pair, $z_i^{t[k]}$ is the identified top $k$ latent dimensions. To remove the effects from these dimensions, we replace the values in the top $k$ dimensions with the average value of the latent variables according to Fong & Vedaldi (2017). In the Figure 3 line chart, our model demonstrates a larger performance change as more important dimensions are removed and

---

[11]We exclude the stopwords and most frequent words appeared in all clusters, such as '*beer*' which occurs in every cluster in the *BeerAdvocate* dataset. The results on the other two datasets, as well as comparison with *StandPrior* are shown in Appendix D.3. The semantically coherence within a cluster are in Appendix D.4.

its changes are more obvious than the two baselines. We further randomly delete $k$ latent dimensions and calculate the relative performance change by subtracting the changes caused by random removal (Table below Figure 3). The relative changes in our model are most predominant, followed by *WassersteinVAE*, while relative changes of *StandPrior* are negative, showing that the random removal of latent dimensions can bring even larger changes to the predominant ones.

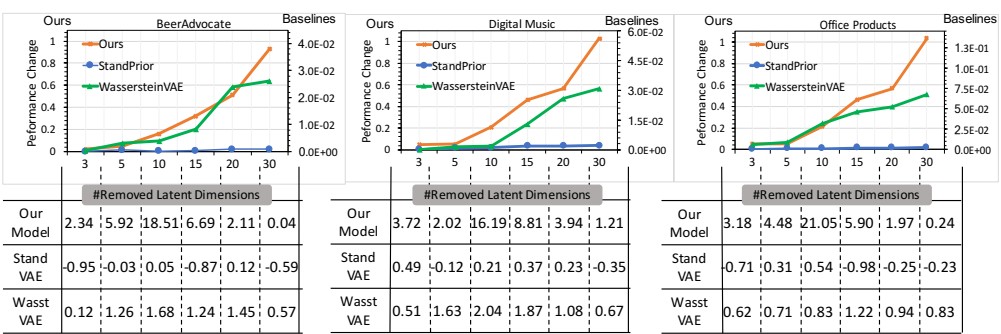

| | 3 | 5 | 10 | 15 | 20 | 30 |
|---|---|---|---|---|---|---|
| **BeerAdvocate** | | | | | | |
| Our Model | 2.34 | 5.92 | 18.51 | 6.69 | 2.11 | 0.04 |
| Stand VAE | -0.95 | -0.03 | 0.05 | -0.87 | 0.12 | -0.59 |
| Wasst VAE | 0.12 | 1.26 | 1.68 | 1.24 | 1.45 | 0.57 |
| **Digital Music** | | | | | | |
| Our Model | 3.72 | 2.02 | 16.19 | 8.81 | 3.94 | 1.21 |
| Stand VAE | 0.49 | -0.12 | 0.21 | 0.37 | 0.23 | -0.35 |
| Wasst VAE | 0.51 | 1.63 | 2.04 | 1.87 | 1.08 | 0.67 |
| **Office Products** | | | | | | |
| Our Model | 3.18 | 4.48 | 21.05 | 5.90 | 1.97 | 0.24 |
| Stand VAE | -0.71 | 0.31 | 0.54 | -0.98 | -0.25 | -0.23 |
| Wasst VAE | 0.62 | 0.71 | 0.83 | 1.22 | 0.94 | 0.83 |

Figure 3: **Top**: The *Comprehensiveness* values by removing the top $k$ most important latent dimensions identified, $k \in \{3, 5, 10, 15, 20, 30\}$. **Bottom Table**: Relative performance changes after subtracting the changes caused by randomly removing $k$ latent dimensions.

### 6.3 CASE STUDY OF GENERATED INTERPRETATIONS

Interpretations for an example user-item pair in *Office Products* are generated by extracting sentences most relevant to user/item latent topics from reviews of like-minded users and past reviews about the item.[12] While existing explainable recommenders are unable to extract information beyond the current user-item review, *GIANT* can extract sentences from reviews of like-minded users to explain the current user-item interaction based on the user/item clustering results. Our human evaluation results presented in Table A7 further show that explanations generated by *GIANT* are better than those from *WassersteinVAE* in terms of relevance, faithfulness and informativeness.

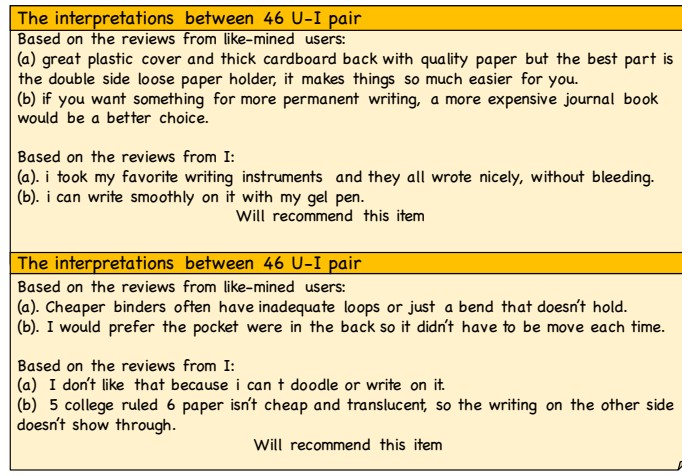

Figure 4: Extracted interpretations from *GIANT* (top) and *WassersteinVAE* (bottom). We first select reviews which are relevant to the user/item assigned cluster, and then extract the most relevant sentences based on their Euclidean distances from the cluster topic representations. Interpretations extracted by *GIANT* appear to be more faithful with the recommender's decision, while those extracted by *Wasserstein-VAE* do not support the recommendation decision.

## 7 CONCLUSION

In this paper, we leverage the user/item clusters sharing common interests/characteristics obtained from the user-item interaction graph to refine the review text latent factors via our proposed geometric information bottleneck (**GIANT**). We empirically show that **GIANT** is better in learning a semantically coherent and interpretable latent space and the generated explanations are more faithful to the model decisions, while achieving comparable rating prediction accuracy on three commonly used datasets.

---

[12]Details of explanation generation can be found in Appendix E.

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

## A EXPERIMENT SETUP

### A.1 DATASET DETAILS AND DATA PROCESSING

We choose three commonly used e-commerce datasets which not only contain the interaction interactions between user and item but also the review texts. Amazon review data [13] is one of the popular dataset collection, consisting of 24 kinds of products. As our method derive the user/item mainly based on their reviews, we use the dense subset, 5-core, of the amazon review dataset that extract the user-item pairs, such that each of the remaining users and items have at least 5 reviews each. Considering the training efficiency and topics should be familiar to general readers, we select the two datasets with around 50k reviews, *Digital Music* and *Office Products*. Besides, we use a sparser dataset *BeerAdvocate* dataset, i.e., the lower rating of observed ratings between total interactions. Table A1 summarizes the statistics of the three datasets. We further filter out users or items with more than 100 reviews/interactions and truncate the reviews to length of [200,400,300] for the three datasets, respectively. We randomly select 80%, 10%, 10% as the train/validation/test sets.

| Datasets | # Interactions | # Users | # Items | # Reviews Per User | # Reviews Per Item | AvgRating | Sparsity |
|---|---|---|---|---|---|---|---|
| *BeerAdvocate* | 35450 | 6939 | 13122 | 56 | 31 | 3.74 | 0.00044 |
| *Digital Music* | 51768 | 5541 | 3568 | 20 | 40 | 4.22 | 0.00294 |
| *Office Products* | 35999 | 4902 | 2364 | 86 | 99 | 4.33 | 0.00349 |

Table A1: Datasets statistics. All the datasets have ratings in the range of 1-5. The *BeerAdvocate* dataset has the smallest sparsity.

### A.2 HYPER-PARAMETER SETTINGS

We use Adam for optimization with the initial learning rate set to 0.001. The batch size is set to 32, 64, and 32, and training epochs are 10, 10, and 15 for the three datasets, respectively. The adjacency matrix in graph is binary: user-item entries with the interacted ratings higher than the average corpus-wide rating is set to 1, and 0 otherwise. We stop training GCN when it reaches the best recall value in the validation set. The cluster centroid vectors are derived by K-means. The graph node feature dimension $d$ is 64, and the number of clusters $K$ are searched in $[16, 32, 64, 128, 256, 512]$. In CNN review encoding, the word embedding and ID embedding are both initialized by the uniform distribution $(-0.1, 0.1)$, their dimensions are set to 300 and 64, respectively. We use two CNN kernels with size 2 and 3, and the dimension of 32 to encode the reviews. As such, the CNN-encoder outputs the review representations with dimension of 64 (i.e., $32 \times 2$). To combine multiple reviews for each user or item, we use the attention mechanism where the attention weights are derived through two consecutive linear transformation layers of $64 \to 32 \to 1$. In **GIANT**, the latent variable dimension is set to the same as the number of clusters $K$. The softmax temperature $\tau$ is searched in $[1, 2, 4, 6, 8]$. The non-linear activation function in both encoder and decoder is ReLU. We use a linear layer with $[K, 64]$ weights to map the the latent variable $z_n^t$ to the same space of the ID features. For $f_{cls}$ used in rating prediction, we first apply the ReLU function on the result of element-wise multiplication of the user and the item features, then apply a linear layer with the shape of $[64, 1]$ to transform the ReLU output to a rating scalar.

## B TRANSFERABLE INFORMATION BOTTLENECK

**Theorem** Suppose we use independent well-trained encoder-decoder based architectures to model the features for different modalities (e.g., graph and text), then a given input $x_n$ has representations $x_n^t \in X^t$ and $x_n^g \in X^g$ under different modalities with the constraint $I(X^t; X^g) \geq I_x$ since $X^t$ and $X^g$ which are from the same input, should be relevant. Also, assuming $I(Z^t; X^t) \leq I_c$ and $I(Z^g; X^g) \leq I_c$, then the following property holds:

$$I(X^t; Z^g) \leq H(X^t) - H(X^g) + H(X^g|X^t) + I_c$$

**Proof:** We first apply the chain rule twice in different orders for three sources $X^t$, $X^g$ and $Z^g$ below:

---

[13]https://jmcauley.ucsd.edu/data/amazon/

$$H(X^t, X^g, Z^g) = H(X^g|X^t, Z^g) + H(X^t|Z^g) + H(Z^g),$$
$$H(X^t, X^g, Z^g) = H(X^t|X^g, Z^g) + H(X^g|Z^g) + H(Z^g).$$

Since the right parts of the above two equations are the same, we have:

$$H(X^g|X^t, Z^g) + H(X^t|Z^g) + H(Z^g) = H(X^t|X^g, Z^g) + H(X^g|Z^g) + H(Z^g),$$

which can be simplified as:

$$H(X^t|Z^g) = H(X^g|Z^g) + H(X^t|X^g, Z^g) - H(X^g|X^t, Z^g).$$

Since $Z^g$, $X^t$ and $X^g$ are derived from the same set of samples but under different modalities or representations, they are obviously not independent. Hence, we have $H(X^g|X^t, Z^g) \leq H(X^g|X^t)$ (as $X^g$ and $Z^g$ are dependent) and $H(X^t|X^g, Z^g) \geq 0$ (as $X^t$ and $X^g$ are dependent). Thus, we can replace $H(X^g|X^t, Z^g)$ with $H(X^g|X^t)$, and replace $H(X^t|X^g, Z^g)$ with 0, to derive the following inequality:

$$H(X^t|Z^g) \geq H(X^g|Z^g) - H(X^g|X^t)$$

By applying the properties of conditional differential entropy, which yields $H(X^g|Z^g) = H(X^g) - I(X^g, Z^g)$, the above formula can be simplified as:

$$H(X^t|Z^g) \geq H(X^g) - I(X^g; Z^g) - H(X^g|X^t)$$

Accordingly, we have:

$$\begin{aligned}
I(X^t; Z^g) &= H(X^t) - H(X^t|Z^g) \\
&\leq H(X^t) - H(X^g) + I(X^g; Z^g) + H(X^g|X^t) \\
&\leq H(X^t) - H(X^g) + H(X^g|X^t) + I_c
\end{aligned} \tag{12}$$

$\square$

Equation 12 shows that the lowest upper bound of the transferable information bottleneck is $I(X^t; Z^g) \leq I_c$ when the distribution of two modalities are the same, $X^t = X^g$, and the conditional entropy of $H(X^g|X^t)$ is 0. Let's take $H = H(X^g) - H(X^t) + H(X^t|X^g)$, then the Equation 12 can be simplified as $I(X^g; Z^t) \leq H + I_c$, where $I_c$ is the upper boundary of information bottleneck for both modalities.

Recall the Lagrange multiplier based objective function defined in Eq. 1,

$$O_{IB} = I(\bar{X}; Z) - \beta \cdot I(X; Z),$$

which aims to optimise:

$$\max_\theta I(\bar{X}; Z) \quad \text{s.t.} \quad I(X; Z)) \leq I_c.$$

Now, let us focus on the optimisation on the encoder-decoder framework in text by denoting the variables with the $t$ superscript. If we use the latent distribution learned from the graph modality, $Z^g$, to impose constraints on $Z^t$, the objective can be rewritten as:

$$\max_\theta I(\bar{X}^t; Z^t) \quad \text{s.t.} \quad I(X^t; Z^g) \leq H + I_c.$$

Since $H = H(X^g) - H(X^t) + H(X^t|X^g)$ is irrelevant with our optimisation, the Lagrange multiplier based objective function can be the rewritten in the form of Eq. 3:

$$O_{IB}^t = I(\bar{X}^t; Z^t) - \beta \cdot I(X^t; Z^g),$$

That is, there is no need to define the prior for the text modality. Instead, we can use the posterior distribution of the latent variable from the graph modality as the regularisation to guide the training on the text data.

## C  Optimising Objective of Transferable Information Bottleneck

**Information Bottleneck:** Recall that the IB objective in this work has the form $I(Z^t; \bar{X}^t) - \beta \cdot I(X^t; Z^g)$, where

$$I(Z^t; \bar{X}^t) = \int p(\bar{x}^t, z^t) \cdot \log \frac{p(\bar{x}^t | z^t)}{p(\bar{x}^t)} d\bar{x}^t dz^t,$$

where $p(\cdot)$ is the true distribution which is not observed. According to the proof given by (Alemi et al., 2017), we consider the posterior estimation $q(\bar{x}_t | z_t)$ in the decoder. According to Gibbs' inequality, we have:

$$I(Z^t; \bar{X}^t) \geq \int p(\bar{x}^t, z^t) \cdot \log \frac{p(\bar{x}^t | z^t)}{q(\bar{x}^t)} d\bar{x}^t dz^t$$

$$= \int p(\bar{x}^t, z^t) \cdot \log p(\bar{x}^t | z^t) d\bar{x}^t dz^t - H(\bar{X}^t).$$

Since the entropy of the decoding results $H(\bar{X}^t)$ is independent of model optimisation, we only need to consider the posterior estimation based lower bound. According to (Alemi et al., 2017), we have the following empirical approximation by reparameterisation trick:

$$\frac{1}{N} \Sigma_{n=1}^N \mathbb{E}_{\epsilon \sim p(\epsilon)} [\log q(\bar{x}_n^t | f(x_n^t, \epsilon)]$$

For the Lagrange multiplied term $I(X^t, Z^g)$ which is different from the standard information bottleneck based $\beta$-VAE, we do consider the information bottleneck of $I(X^t, Z^t)$ first, because $I(X^t, Z^g) \leq H + I(X^t, Z^t)$ according to the proof given in section B. Here, we have the following bound by applying Gibbs' inequality:

$$I(X^t; Z^t) = \int p(x^t, z^t) \log p(z^t | x^t) dx^t dz^t - \int p(z^t) \log p(z^t) dz^t.$$

Since estimating the prior distribution of $Z_t$ might be difficult, based on the definition of transferable information bottleneck, we apply the Gibbs' inequality and have:

$$I(X^t; Z^g) \leq \int p(z^t | x^t) p(x^t) \log \frac{p(z^t | x^t)}{r(z^g)} dx^t dz^t + H,$$

where $r(z^g)$ is the posterior distribution from the graph-based latent representation, and $H$ is decided by the prior of two modalities which is independent of model optimisation. By combining the two results, the above bound can be approximated by the reparameterisation trick (Kingma & Welling, 2013) with a Gaussian random variable $\epsilon$:

$$O_{IB} = \Sigma_{n=1}^N \mathbb{E}_{\epsilon \sim p(\epsilon)} [- \underbrace{\log q(\bar{x}_n^t | f(x_n^t, \epsilon)]}_{\text{Recons. term: Q(x}_n^t)} / N + \beta \cdot \underbrace{\text{KL}(p(Z^t | x_n^t, \epsilon) | r(Z^g))}_{\text{KL-div. term: KL(Z}^t | Z^g)},$$

**Further discussion about the reconstruction term $Q(x_n^t)$**

In section 4.2, we propose to use the Nadaraya-Watson estimator (Hall et al., 1999) to approximate the conditional probability $q(\bar{x}_n^t | f(x_n^t, \epsilon)]$. The idea is to insert the decoding result $\bar{x}_n^t$ to the space of training samples $\{x_n^t\}$, and take the kernel density estimation method to approximate $Q(x_n^t)$ by Bayesian rule:

$$Q(x_n^t) = \log \frac{\hat{p}(\bar{x}_{n,\epsilon}^t, x_n^t)}{\hat{p}(x_n^t)} = \log \frac{\frac{1}{N} \sum_{j=1}^N \kappa(\frac{x_n^t - x_j^t}{h}) \cdot \kappa(\frac{\bar{x}_{n,\epsilon}^t - x_j^t}{h})}{\frac{1}{N} \sum_{j=1}^N \kappa(\frac{x_n^t - x_j^t}{h})},$$

Since the input feature for $x_n^t$ is fixed, we only care about the updating of reconstruction through encoder-decoder architecture. Thus we have:

$$Q(x_n^t) \propto \frac{1}{N} \sum_{j=1}^{N} \kappa(\frac{x_n^t - x_j^t}{h}) \cdot \kappa(\frac{\bar{x}_{n,\epsilon}^t - x_j^t}{h})$$

To simplify the above estimation, we apply the RBF-kernel and triangle inequality in estimation and have:

$$
\begin{aligned}
Q(x_n^t) &\propto \log \frac{1}{N} \sum_{j=1}^{N} \kappa(\frac{x_n^t - x_j^t}{h}) \cdot \kappa(\frac{\bar{x}_{n,\epsilon}^t - x_j^t}{h}) \\
&= \log \frac{1}{N} \sum_{j=1}^{N} \exp(-||x_n^t - x_j^t||^2) \cdot \exp(-||\bar{x}_{n,\epsilon}^t - x_j^t||^2) \\
&= \log \frac{1}{N} \sum_{j=1}^{N} \exp(-||x_n^t - x_j^t||^2 - ||\bar{x}_{n,\epsilon}^t - x_j^t||^2) \\
&\leq \log \frac{1}{N} \sum_{j=1}^{N} \exp(-||\bar{x}_{n,\epsilon}^t - x_n^t||^2) \\
&= \log[\exp(-||\bar{x}_{n,\epsilon}^t - x_n^t||^2)]
\end{aligned}
$$

Thus, we are able to optimise the $Q(x_n^t)$ by mean square error if we choose the natural logarithm function:

$$-Q(x_n^t) \geq -\ln(\exp(-||\bar{x}_{n,\epsilon}^t - x_n^t||^2)) = ||\bar{x}_{n,\epsilon}^t - x_n^t||^2$$

Therefore, we can rewrite the lower boundary as well as the learning objective $O_{IB}$ by:

$$O_{IB} = \frac{1}{N} \Sigma_{n=1}^{N} \mathbb{E}_{\epsilon \sim p(\epsilon)}[||\bar{x}_{n,\epsilon}^t - x_n^t||^2] + \beta \cdot \mathrm{KL}(p(Z^t|x_n^t, \epsilon)|r(Z^g)),$$

# D ADDITIONAL EXPERIMENTAL RESULTS

## D.1 EVALUATION RESULTS ON RANKING-BASED METRICS

In addition to the accuracy-based metrics, i.e., MSE and MAE, we also evaluate models using the ranking-based metrics, i.e., Precision and Recall (See Table A2). We observe that **GIANT** achieves the best precision results in general across all the datasets, with a more prominent improvement compared to baselines on *Office Products*. In terms of recall, **GIANT** outperforms the others on *Digital Music* and *Office Products*, but it is inferior on *BeerAdvocate*. *WasersteinVAE* appears to be the second-best model. The results show that **GIANT** is more effective on datasets covering products with distinct features, but its advantage over the baselines is less obvious on datasets with similar feature descriptions, such as *BeerAdvocate*.

$$
\begin{aligned}
\text{Precision} &= \frac{\#\text{of recommended items @k that are relevant}}{\#\text{recommended items}} \\
\text{Recall} &= \frac{\#\text{of recommended items @k that are relevant}}{\#\text{relevant items}}
\end{aligned}
$$

## D.2 THE IMPACT OF CLUSTER NUMBER

We show the the impact of different number of clusters $K$, i.e., the dimensionality of latent variable $z_n$, on review rating prediction results in Table A3. The best rating prediction results are obtained for *Digital Music* and *Office Products* when the number of clusters is set to 128. But on *BeerAdvocate*, the optimal cluster number is 256.

| Models | | NARRE | AutoEncoder | WassasteinVAE | StandPrior | GIANT |
|---|---|---|---|---|---|---|
| **BeerAdvocate** | Precision@1 | 0.24 | 0.16 | 0.20 | 0.20 | 0.24 |
| | Recall@1 | 0.20 | 0.15 | 0.17 | 0.16 | 0.15 |
| **Digital Music** | Precision@1 | 0.20 | 0.16 | 0.22 | 0.23 | 0.23 |
| | Recall@1 | 0.21 | 0.17 | 0.24 | 0.23 | 0.23 |
| **Office Products** | Precision@1 | 0.24 | 0.20 | 0.29 | 0.28 | 0.31 |
| | Recall@1 | 0.22 | 0.18 | 0.23 | 0.19 | 0.23 |

Table A2: The rank-based evaluation results, i.e., Precision and Recall for all the baseline and our proposed **GIANT**.

| Dataset | 16 | 32 | 64 | 128 | 256 | 512 |
|---|---|---|---|---|---|---|
| **BeerAdvocate** | 75.65 | 75.56 | 75.53 | 75.46 | **75.36** | 75.74 |
| **Digital Music** | 93.51 | 93.24 | 92.93 | **92.87** | 93.33 | 93.10 |
| **Office Products** | 84.78 | 84.53 | 84.42 | **84.32** | 84.35 | 84.48 |

Table A3: **GINAT** rating prediction performance in Mean Square Error (MSE %) with different number of clusters.

## D.3 CLUSTER SEPARABILITY

As each dimension of the latent variables $z_n$ corresponds to a cluster, for the $k$-th dimension, we can search for its relevant reviews which have the highest value in $z_k$ and list the most frequent words in the review set as the representative topic words. We identify '*ABBA*' and their popular songs, computer peripherals such as '*network cable*' and '*router*' in *Digital Music* and *Office Products* datasets, respectively.

| | GIANT | WassersteinVAE |
|---|---|---|
| *Digital Music* | **dr dre**, **prince**, **used known**, west coast, mirrors, love, hate, westside story, jadekiss, always | quot, album, **Ask Rufus**, song, **funk**, **voice**, music, band, soul, love |
| | **bangles**, **vicki**, **vocal**, place, live, hero, takes, fall, liverpool, beatles | song, **everglow**, album, band, **beautiful**, track, **lyrics**, suspension, record, Christian, |
| | **dancing queen**, **take chance**, **mamma mia**, abba, greatest hit, money money, gimme gimme, super trouper, knowing knowing, gold | record, album, song, band, track, time, sound, **head heart**, **folk**, **live**, love |
| *Office Products* | **set** , **computer**, **wireless**, network, cable, router, edit, pencil, ink | **scan**, **software**, pencil, **Mac OS**, work, printer, small, printed, color, file |
| | **photo**, **desk**, **laser**, scanner, work, scan, epson, scaning, enough, easy | easy use, **dry erase**, folder, **mouse pad**, office, **ink cartridge**, works, used, post note |
| | pencil, **tape**, **ink**, pen, anyway, pen, **rubber**, clip, pretty, eraser | **recommend**, printer, folder, **paper**, easy use, **boxes**, feature, stapler, pencil, canon, color |

Table A4: The most prominent words (sorted by occurrence frequency) in three randomly selected clusters from **GIANT** and *WassersteinVAE* on *Digital Music* and *Office Products* datasets. For each dataset, we highlight the top 3 words that are not found in the other two clusters. **GIANT** generates better separable topics while *WassersteinVAE* fail to generate clear topic pattern.

We also compare with the generated clusters using *StandPrior*. The results are shown in Table A5. It is difficult to see a clear topic pattern in *StandPrior* as the top words largely overlap in different clusters. The first several words are all '*hop*', '*malt*' and '*good*'. The results are even worse than *WassersteinVAE*, which can at least generate diverse clusters, reflected in the distinguished words in each clusters.

| | |
|---|---|
| *BeerAdvocate* | hop, good, malt, aroma, **hint**, pour, **great**, well, mouthfeel, much |
| | hop, aroma, good, sweet, light, malt, finish, **brew**, poured, **glass** |
| | hop, malt, light, sweet, much, **carbonation**, good, well, mouthfeel, pour |
| *Digital Music* | great, best, band, love, **hit**, well, first, make, lyric, come |
| | love, great, best, band, **sound**, well, first, even, make, lyric |
| | great, love, best, well, band, lyric, make, fan, first |
| *Office Products* | use, work, well, binder, great, will, **easy**, **nice**, label, **ink** |
| | label, easy, tape, color, great, using, work, make, well, really |
| | use, label, product, tape, color, binder, **quality**, really, **scanner**, great |

Table A5: The most prominent words (sorted by occurrence frequency) in three randomly selected clusters from *StandPrior* on *BeerAdvocte*, *Digital Music* and *Office Products* datasets. For each dataset, we highlight the top 3 words that are not found in the other two clusters. *StandPrior* produces topics which contain largely overlapped words.

| Models | BeerAdvocate | | Digital Music | | Office Products | |
|---|---|---|---|---|---|---|
| | User | Item | User | Item | User | Item |
| *Graph* | 0.092 | 0.085 | 0.155 | 0.214 | 0.245 | 0.200 |
| *GIANT* | 0.490 | 0.481 | 0.457 | 0.471 | 0.492 | 0.407 |

Table A6: Average cosine similarity of review pairs within each cluster as the coherence measure. A larger similarity value means a better coherence. **GIANT** generates significantly more coherent clusters than graph clusters.

### D.4 CLUSTER COHERENCE

We explore the difference between our generated clusters $z_{nk}$ and the clusters derived in the graph. To verify the capability of creating *semantically coherent clusters*, we propose to measure the cluster coherence as the average cosine similarity between every review document pair within a cluster. We first obtain the document-level review representations by feeding the reviews from the test set to our pre-trained CNN encoder. We then calculate the average cosine similarity between the representations of each review pair in a cluster, and are further averaged across all clusters (in Table A6).

## E HUMAN EVALUATION FOR INTERPRETABLITY

We conduct human evaluation to validate the interpretablity of our proposed method. To make it easier for humans to understand the rationales behind model decisions, we extract the most relevant sentences from user/item reviews as interpretation for a specific user-item pair (user $u$ and item $i$). Example generated interpretations are illustrated in Figure 1 and Figure 4.

In particular, our *GIANT* model infers the latent topic for user $u$ and item $i$, respectively. Such a topic essentially indicates which cluster user $u$ or item $i$ belongs to. We can then identify the user candidate reviews as the past reviews on item $i$ written by the users (including $u$) in the same cluster as $u$. Similarly, we identify item candidate reviews from the past reviews on the item $i$ which have their most prominent topic the same as the item latent topic.

Afterwards, we represent each topic by its top-associated 5 words and derive the topic representation by the aggregated word-level GloVe word embeddings. [14] These words are selected based on the TFIDF scores of all words of reviews in the same topic cluster, with stop words filtered. The review sentence representations are also derived based on the aggregated constituent word GloVe embeddings. The most relevant sentences from user candidate reviews can then be extracted as a summary of the user $u$'s preferences based on their cosine similarity with the user latent topic representation. Similarly, the most relevant sentences from item candidate reviews are extracted as a summary of item $i$'s characteristics based on their cosine similarity with the item latent topic representation. Apart from the user $u$'s preferences and item $i$'s characteristics, we also present to human evaluators the model's recommendation suggestion as *will recommend*, if the model's predicted rating score is above the average predicted rating; or *won't recommend*, otherwise.

---

[14]We also experimented with more representative words, but observed less discriminated topic clusters.

We propose three evaluation metrics, *relevance*, *faithfulness* and *informativeness* and ask three English-proficient human evaluators to give 1-5 score to the generated interpretations from *WassersteinVAE* and *GIANT* on 120 randomly selected user-item pairs from the *Office Products* [15].

- **Relevance**. If the extracted sentences from user reviews and item reviews are about the relevant topic/aspect/subject? A higher score should be given to the interpretation with more overlapping aspects.

- **Faithfulness**. Do the extracted sentences from user reviews and item reviews lead to the model's recommendation suggestions?

- **Informativeness**. Do the given interpretations capture the user preferences and item characteristics well?

| Metric | Relevance | Faithfulness | Informativeness |
|---|---|---|---|
| *WassersteinVAE* | 3.54 | 2.84 | 3.84 |
| *GIANT* | 3.77 | 3.27 | 4.17 |

Table A7: Human evaluation results on *relevance*, *faithfulness* and *informativeness* for the generated interpretations of randomly selected 120 user-item pairs from the *Office Products* dataset.

From the results shown in Table A7, we can observe that interpretations generated by our model are better compared to *WassersteinVAE* across all three aspects.

---

[15]Evaluation on the Beer and Music products appears to be more difficult as it requires prior knowledge on the specific products. We will leave it to future work.

