# OpenReview forum: "Explainable Recommender with Geometric Information Bottleneck"
_ICLR.cc/2023/Conference — Submitted to ICLR 2023_

### Official Review · Reviewer_fUWF · 2022-10-19

**Confidence:** 3
**Correctness:** 4
**Technical Novelty And Significance:** 3
**Empirical Novelty And Significance:** 3
**Recommendation:** 5

**Clarity, Quality, Novelty And Reproducibility:**

The paper is rather clear and well written.
The combination of all the blocks seems to be novel. The use of VAE or user-item interaction graphs to learn embeddings in recommender systems is not new. What is novel here is the proposition of regularization terms from graph modularity to text modularity. The proposed prior for VAE is more sophisticated than for StandPrior.
Experiments have been made on publicly available benchmarks and the authors provide the set of hyperparameters and chosen optimizer to ensure reproducibility of the results.


**Strength And Weaknesses:**

Strengths:

The paper is quite clear and enjoyable to read.
The approach is clearly stated with the different steps:
-interaction graph computation
-encoding of the nodes
-clustering
-VAE: prior / posterior
-Final objective function

The framework is supported by multiple experiments on 3 public datasets to evaluate quantitatively accuracy of the recommendation but also importance of each block of the loss by appropriate ablation study. Finally there is also a quantitative analysis of the interpretability of the recommendations based on the cluster analysis: as each dimension of the latent variables from the VAE encoding part are associated to a cluster, we can interpret the recommendation of a given item by looking at the the most frequent words associated to reviews from clusters associated to the highest values of coefficients of the latent vector of the item.

Weaknesses / questions:

I have several questions:

Q1: Why not compare GIANT with Wasserstein VAE and only StandPrior for interpretability evaluation?

Q2: What is the architecture of the predict head? Simply dot product of the latent vectors from the VAE?

Q3: How are the text reviews treated before LightGCN? Is it the same treatment as for the input of the VAE encoder?

Q4: Eq. (1). Which value do you take in practice for $I_c$?

Q5: Is K-means applied on all types of nodes at the same time or two applications on user and items separately? I think there is only one application but p.4 the differentiation of $C^u_k$ and $C^i_k$ is a bit confusing.

Q6: None of the methods described in the Related work section for explainable recommenders have been tested against GIANT. Am I mistaken?

Q7: Did you try the Wassertein metric instead of KL divergence to calculate the distribution discrepancy between posterior and prior? (to make WassersteinVAE benefit from a better prior).

Minor (typos):
-p.3 text before equation 1, extra comma to remove after decode arrow
-p.16 missing space between citation and Nadaraya-Watson estimator + bracket typo is conditional probability


**Summary Of The Paper:**

The authors propose a new framework called GIANT (for Geometric InformAtioN boTtleneck) to provide results’ interpretations/explanations of a collaborative -filtering-based recommendation system.

This framework requires first the computation of a user-item interaction graph.
Each node is represented by LightGCN encoding of:
- for a user node: all text reviews the user wrote,
- for an item node: all text reviews posted for this item.

User and item nodes are then clustered via K-means based on Radial Basis Function kernel for distance. The distance between a user or an item to the cluster centroid vectors is used as the probability to be assigned (soft assignment) to the cluster and the corresponding distribution is used as prior to learn user and item latent vectors from a Variational AutoEncoder (VAE).

For a pair of user-item taken from the interaction graph, user and item representations are fed to a VAE trained so that obtained latent vectors of the pair are used to predict the rating from that user for this specific item.

Final loss of the network does not take into account only the regression loss (for rating prediction) but also regularization terms:
-one for constraining the learning of the latent vector in text modality by imposing the prior knowledge from the graph modality (information bottleneck)
-one for preventing learned distribution of latent vector to be chaotic (prior-centralisation)


**Summary Of The Review:**

The approach is showing on the 3 public benchmarks best performance so the method seems promising. Moreover the qualitative evaluation shows indeed interpretation associated is more valuable than the one associated to StandPrior. Why not compare with WassersteinVAE which was giving the second best performance in terms of rating prediction?
My score is currently a weak reject, score that can be increased when my questions (in weaknesses part of the review) are answered.

---

> ### Author Response · Authors · 2022-11-18
> **Response to Reviewer fUWF**
>
> Q1: Why not compare GIANT with Wasserstein VAE and only StandPrior for interpretability evaluation?
>
> R1: Figure 2 shows the rating prediction results of the *Wasserstein distance*, *StandPrior* using KL divergence and our GIANT, as well as two variants using different priors. We added the results of Wasserstein VAE throughout Section 6, including cluster separability, completeness, and generated explanations from GIANT and Wasserstein VAE for human evaluation in Appendix E.
>
> Q2: What is the architecture of the predicted head? Simply dot product of the latent vectors from the VAE?
>
> R2: See the Rating prediction and final objective paragraph in Section 4 and the updated Hyperparameter settings in Appendix A.2. Firstly, we incorporate the ID information (i.e., $\epsilon_{i}$ and $\epsilon_{u}$) with the latent vectors to derive the user/item features. Then, we apply the ReLU function to the element-wise multiplication results of the user and the item features and derive the combined features. A linear layer in the shape of $[64,1]$ is used to transform the combined features into a basic rating scalar.  We finally add the bias from the user and item side (i.e., the $b_{u}$ and $b_{i}$), respectively to derive the final rating.
>
> Q3: How are the text reviews treated before LightGCN? Is it the same treatment as for the input of the VAE encoder?
>
> R3: We only feed the text reviews to the VAE encoder. The input to the LightGCN is the binary interaction matrix with the size of $|U| \times |I|$, with the value of 1 representing positive feedback of a user on an item, and 0 for negative feedback.
>
> Q4: Eq. (1). Which value do you take in practice for $I_c$ ?
>
> R4: $I_c$ denotes the upper bound of the mutual information between $X$ and $Z$, we don't derive a specific value for it. Instead, we give a theory proof about leveraging prior from another modality to build the information bottleneck can also maintain an upper bound for the corresponding mutual information, which is the theory guarantee for our multi-modality variational recommender. Please refer to the updated section 3 for details.
>
> Q5: Is K-means applied on all types of nodes at the same time or two applications on user and items separately? I think there is only one application but p.4 the differentiation of  $C_{k_u}$ and $C_{k_i}$ is a bit confusing.
>
> R5: We apply K-means to user and item nodes, separately, so that user clusters capture users with similar interests while item clusters encode items with similar characteristics.
>
> Q6: None of the methods described in the Related work section for explainable recommenders have been tested against GIANT. Am I mistaken?
>
> R6: Existing explainable recommenders require either human-annotated text spans as explanations or aspects annotated for training aspect-opinion extraction models. They are not directly comparable here since the only supervision in our setup is document-level rating scores.
>
> Q7: Minor (typos): -p.3 text before equation 1, extra comma to remove after decoding arrow -p.16 missing space between citation and Nadaraya-Watson estimator + bracket typo is conditional probability.
>
> R7: Corrected.

---

> > ### Comment · Reviewer_fUWF · 2022-12-02
> > **Thank you very much to the authors for their answers to our reviews and for improving the paper during the rebuttal period**
> >
> > Thank you very much to the authors for their answers to our reviews and for improving the paper during the rebuttal period. The modifications bring valuable content and clarification. I read also carefully the other reviews and the corresponding answers. However, I keep my recommendation "marginally below the acceptance threshold" since my understanding for now is that not 100% of reviewers' concerns are addressed.

---

> > > ### Author Response · Authors · 2022-12-10
> > > **Further response has been posted. Please check**
> > >
> > > We have posted further response to other reviewers' comments after you last comment. In particular, the following have been added after your last comment:
> > >
> > > To Reviewer JQQm:
> > > (1) We added human evaluation results on all three datasets and provided precision@k and recall@k results for the top 10 retrieved items.
> > > (2) We have provided detailed explanation for the equation derivations and questions about priors.
> > >
> > > To Reviewer Scqk:
> > > (1) We have added a summary of retrieval-based explainable recommender systems.
> > > (2) We have provide detailed explanation of step-by-step derivation of our equations.

---

### Official Review · Reviewer_Scqk · 2022-10-20

**Confidence:** 3
**Correctness:** 2
**Technical Novelty And Significance:** 3
**Empirical Novelty And Significance:** Not applicable
**Recommendation:** 5

**Clarity, Quality, Novelty And Reproducibility:**

Clarity, quality and reproducibility could be improved (see weaknesses 1 and 3).
Novelty is okay (see strength 1).

**Strength And Weaknesses:**

Strength:
1. The overall idea (use mutual information to connect multi-modal data) is interesting and novel, and it could be useful for explainable recommendation.
2. The introduction includes a good summarization of existing issues in explainable recommendation, especially that the reviews usually are not good ground-truth labels for explanations, and that getting the true ground-truth is very difficult.

Weakness:
1. The major weakness is about the explanation introduction, quality, and evaluation.
(1) Method: I cannot find a clear and formal introduction about how explanations are generated in the method section. The introduciton seems to talk about explanation generation with one sentence: "and we extract the representative words in each dimension to offer recommendation explanations." According to this sentence and also Fig. 4, I think the explanations are actually many words in some clusters that are activated for a user-item pair. This is different from the example in Fig. 1, which contain only a few words. I suggest the authors more clearly introduce their explanation generation method, since explainability is a core contribution here.
(2) Explanation quality: When borrowing words from similar items/users, how can you ensure that the selected words are relevant to the item and faithful (not misleading)? Also, the current methods only ensures similarity between input and decoded embeddings. Can this ensure that the decoded words are still relevant with the input words? Even if we can correctly decode a review, how can we ensure that the decoded words serve as good explanations? Can you help explain more about these questions?
(3) Evaluation: No formal comparison with baselines in terms of explanation quality (e.g., whether the explanations are correct or useful). For example, what are the example explanations that you will show to users, and any comparions with baselines to prove your explanation quality? Also, any ablation study on whether the regularization really helps in improving explanation quality?
2. Some descriptions about related works are not rigorous
(1) "However, the aforementioned approaches suffer from two major
issues: they rely heavily on the accuracy of sentiment analysis tools (Guan et al., 2018); they tend
to ignore implicit interactions not expressed directly in the individual review text." I do not think all mentioned related works rely on sentiment analysis tools? What do you mean by ignore implicit interactions? I think most methods consider user-item interaction information in the model (e.g., as model outputs)?
(2) NAREE should be NARRE.
3. The theory part is unclear to me.
(1) Why the second term in Eq. (3) is I(X_t, Z_g) rather than I(X_t, Z_t)? Why the first term considers only Z_t but not Z_g? Do you use Eq. (2) to dervie Eq. (3)? Can you give a step-by-step derivation here? For example, can you first replace X, Z in Eq. (1) with Xt, Xg and Zt, Zg and gradually derive each equation?
(2) The proof in Section B is also difficult to understand. I am an expert in explainable recommendation but I am not an expert in information bottleneck. Every derivation step needs to be clearly written for me to carefully check the correctness of the proof, e.g., what is the chain rule, how you apply it twice.

**Summary Of The Paper:**

This paper presents a review-based explainable recommedation model that predicts user preference and gives explanations based on user/item clusters. The core idea is to learn two sets of clusters, one based on text reviews and the other based on user/item interaction information. This explicitly ensures that information about similar users from the aspect of collaborative filtering can facilitate explanation generation for each other. Mathematically, this is achieved by using information bottleneck: the authors maximize the information between two sets of clusters, while ensuring compression (deriving compact clusters) and reconstruction also through information-based perspective.
Contribution:
1. The idea to build a connection between multi-modal data (text and interaction) through mutual information is quite interesting and is novel to explainable recommendation according to my knowledge.
2. The summarization of existing issues in explainable recommendation is good, especially the difficulty in getting ground-truth labels for explanations and the low quality for reviews.

**Summary Of The Review:**

The idea is novel and interesting, and the summarization about current issues in explainable recommendation also provides interesting insights. However, the explanation generation method is not clearly introduced and discussed, and the explanation quality is not well evaluated. Moreover, the related theories and equations can be better introduced to ensure readability. In its current state, it is difficult for me to judge whether the method has issues.

---

> ### Author Response · Authors · 2022-11-18
> **Response to Reviewer Scqk: About Interpretation and Related Work**
>
> **About Interpretation**:
>
> Q1. I suggest the authors clearly introduce their explanation generation method since explainability is a core contribution here.
>
> R1: The interpretations are the extracted representative words in each cluster in the initial submission. To save space, we only showed a few words in Fig 1. In the revised paper, we extract sentences from user/item reviews which are most relevant to the learned latent topic, i.e., the representative words, for the user and item, respectively. Please refer to the updated Fig1 and Fig 4 in Section 6.3 case study for the generated interpretation examples, and refer to Appendix E for the model interpretation generation process.
>
> R2. Explanation quality: When borrowing words from similar items/users, how can you ensure that the selected words are relevant to the item and faithful (not misleading)?
>
> A2: Please refer to the explanation generation process in Appendix E and our human evaluation results in terms of relevance, faithfulness and and informativeness.
>
> Q3.  The current methods only ensure similarity between input and decoded embeddings. Can this ensure that the decoded words are still relevant to the input words? Even if we can correctly decode a review, how can we ensure that the decoded words serve as good explanations? Can you help explain more about these questions?
>
> R3: Our explanations are NOT generated from the decoded words. After our encoder-decoder framework is trained, we can feed a review x to derive the latent variable z in k dimensions, which can be considered as the k-cluster assignment distribution. All the user/item reviews can be thus assigned  into various clusters. For each topic cluster, the top words are extracted based on their TFIDF values from all reviews in the cluster. Sentence-level explanations are generated based on the process described in Appendix E.
>
> Q4: No formal comparison with baselines in terms of explanation quality (e.g., whether the explanations are correct or useful). For example, what are the example explanations that you will show to users, and any comparisons with baselines to prove your explanation quality?
>
> R4: See the updated Fig1 and Fig 4 in Section 6.3 case study for the generated interpretation examples, which are in the same form as the evaluation examples shown to our human evaluators.  Refer to Appendix E for the model interpretation generation process, as well as the human evaluation results in Table A7, in Appendix E.
>
> Q5: Any ablation study on whether the regularization really helps in improving explanation quality?
>
> R5: See the general response to all the reviewers. The quality of generated interpretations are closely related to the generated clusters, i.e., the latent variables, as we use the relatedness between selected reviews and the extracted representative words in the cluster. To evaluate the latent variables, we have shown results in semantic coherence within clusters, separability across clusters, and faithfulness evaluation showing the superiority of our proposed regulariser by comparing GIANT with the other VAE-based baselines, StandPrior and Wasserstein. In this way, we verified the superiority of our proposed Geometric prior.
>
> **About Related work**:
>
> Q1: Some descriptions about related works are not rigorous.
>
> R1:  We agree that not all the related work relies on sentiment tools. Please refer to the updated section. Some existing explainable recommenders require extracting the aspects mentioned in reviews. They either rely on aspect extractor tools, e.g., Sentries (https://github.com/evison/Sentires) or a sentiment lexicon built on aspects as in *Justifying Recommendations using Distantly-Labeled Reviews and Fine-Grained Aspects*.  Other explainable recommenders rely on human-annotated rationales to generate review-level explanations such as tips (*Neural Rating Regression with Abstractive Tips Generation for Recommendation*).
> Here, implicit interactions are in contrast to the explicit/literal text spans in the given reviews. We have revised this part to avoid misunderstanding. Please refer to revised Section 2.
>
> Q2: NAREE should be NARRE.
>
> R2: Corrected.

---

> > ### Comment · Reviewer_Scqk · 2022-11-21
> > **Current Concerns about Explanation Generation and Evaluation**
> >
> > Thank the authors for carefully addressing my concerns and refining the explanation generation method. This part has been improved. My current concern is that the new method and evaluation may still has problems.
> >
> > 1. The authors said in Apendix E that "We can then identify the user candidate reviews as the past reviews on item i written by the users (including u) in the same cluster as u"
> > Do the authors use the review for item i written by user u when generating the explanation for (u, i)? This is a leak of information, as this review is usually close to ground-truth label for both rating and explanation. We usually remove such information from the training data in explainable recommendation. If the authors insist that their explanation is even better, they may include this review (written by user u for item i) as a baseline in the human evaluation to see the result.
> >
> > 2. In the human evaluation, the authors only consider one baseline, Wasserstein VAE. Is it a widely-used or SOTA method for generating explanations in recommeder systems? Why not include more, e.g., NARRE?

---

> > > ### Author Response · Authors · 2022-11-21
> > > **Response to "Current Concerns about Explanation Generation and Evaluation"**
> > >
> > > Q1: Do the authors use the review for item i written by user u when generating the explanation for (u, i). If the authors insist that their explanation is even better, they may include this review (written by user u for item i) as a baseline in the human evaluation to see the result.
> > >
> > > R1: We do not use the (u, i) review to generate the explanation for (u, i), also do not use it in training. In our experiment setup, we remove those (u,i) pairs from training data which have appeared in testing data to avoid information leaking.
> > > To simulate a real situation, the recommendation of item i to user u occurs without seeing the review for item i written by user u. Including the (u,i) reviews as a baseline in human evaluation will lead to evaluation bias.
> > >
> > > Q2: Why not include more baselines for interpretation evaluation?
> > >
> > > R2: Please see the response to Reviewer fUWF Q6. Existing explainable recommenders require either human-annotated text spans as explanations or aspects annotated for training aspect-opinion extraction models. They are not directly comparable here since the only supervision in our setup is document-level rating scores.  Here, we choose the Wasserstein VAE as a strong baseline since it is a strong generative model and addresses the same issue of unstable training progress due to inappropriate prior. It is also the best-performing model among all the baselines we experimented with in terms of review rating prediction, and quantitative evaluation of explanations generated.

---

> > > > ### Comment · Reviewer_Scqk · 2022-11-22
> > > > **Still Confused**
> > > >
> > > > 1. According to the following sentence, you include the review on item i written by user u in the candidate reviews. Is my understanding correct? If yes, could you let me know why you think this is not leak of data? If no, could you explain what do you really mean by the next sentence?
> > > > "We can then identify the user candidate reviews as the past reviews on item i written by the users (including u) in the same cluster as u"
> > > >
> > > > 2. Many explainable recommendation methods such as [1][2] also do not require human-annotated text spans or aspects annotated. Actually all retrieval-based method do not require human annotations. Only aspect-based explanations or methods that generate explanations word by word require such annotations. Please also note that the data the authors use (user, item reviews and ratings) also have no significante difference compared with existing review-based methods.
> > > > [1] Wang X, Chen Y, Yang J, Wu L, Wu Z, Xie X. A reinforcement learning framework for explainable recommendation. In2018 IEEE international conference on data mining (ICDM) 2018 Nov 17 (pp. 587-596). IEEE.
> > > > [2] Chen C, Zhang M, Liu Y, Ma S. Neural attentional rating regression with review-level explanations. InProceedings of the 2018 World Wide Web Conference 2018 Apr 10 (pp. 1583-1592).

---

> > > > > ### Author Response · Authors · 2022-11-22
> > > > > **Further discussion about interpretation generation and evaluation**
> > > > >
> > > > > Q1:  Do you include the review on item i written by user u in the candidate reviews. Please explain the sentence ""We can then identify the user candidate reviews as the past reviews on item i written by the users (including u) in the same cluster as u"
> > > > >
> > > > > R1: Note that our model is trained for review rating prediction. Each instance is a review document. When performing rating prediction for a review d written by user u on item i, d is excluded from the training set. However, it might be the case that user u has written a review on item i before, i.e., user u has written multiple reviews on item i.  That is why we “identify the user candidate reviews as the PAST reviews on item i”.  It is however worth noting that such cases (user i has multiple reviews associated with item i) are rare in our datasets. The number of multiple (u, i) reviews in train/test is shown in the table below:
> > > > >
> > > > > | \#duplicate\#total | BeerAdvocate |DigitalMusic | Office Products|
> > > > > | :---------- | ----------- |----------- |----------- |
> > > > > |  **Train**    | 64/28360  | 0/41414| 0/28799 |
> > > > > | **Test**   | 2/3545 | 0/5177 | 0/3600 |
> > > > >
> > > > > The table clearly demonstrates that multiple (u,i) reviews only happen in the domain of *Beer*, and never appear in the other two domains of  *Music* and *Office*.
> > > > >
> > > > > Q2: Comparison with retrieval-based explainable recommender.
> > > > >
> > > > > R2: It is worth noting that the above two retrieval-based methods can extract sentences from item reviews only as explanations, and there is no guarantee that the selected user's past reviews are related to the item.  But in our setup, our model can simultaneously extract sentences from reviews of like-minded users and of similar items, essentially characterising both user preferences and item characteristics for better explanation of recommender decisions. In this regard, none of the cited papers is comparable here. Nevertheless, we have also generated explanations by retrieving sentences from both user and item reviews using the approach in [2] (The paper [1] did not release their source code and hence we were not able to conduct experiments using their approach). Example results on *Office Products* are shown below:
> > > > > ```
> > > > > Based on the user past reviews:
> > > > > (a). The sticky paper can stick to the desktop when you need one hand to write a note, and the other hand for the phone.
> > > > > (b). This is not only a pretty large piece but holds the post it notes inside for easy capture.
> > > > >
> > > > > Based on the item past reviews:
> > > > > (a). This printer performs well, is fast and print quality is very good.
> > > > > (b). Prints great. It wasn’t much pain to set up. To make things easier, I wish it could have bluetooth printing capability.
> > > > >
> > > > > Will recommend the item
> > > > > ```
> > > > >
> > > > > Noted that in the released code of [2], all the input reviews are padded as [ReviewNum, ReviewLen] and the users/items have N reviews, less than ReviewNum, will be padded with (ReviewNum-N) reviews.  The authors did not exclude the (ReviewNum-N) reviews when assigning review-level attention. We found lots of [PAD] reviews are assigned with highest attention weight. To avoid this issue, we firstly remove the [PAD] reviews and select from the remaining reviews.
> > > > >
> > > > > We will update the Human Evaluation results on NARRE later.

---

> > > > > > ### Comment · Reviewer_Scqk · 2022-11-23
> > > > > > **Comments about training details and evaluation**
> > > > > >
> > > > > > 1. As shown by the statistics, it is very rare that a user writes multiple reviews about an item, so the original sentence is quite confusing. Hence I find it strange that the authors think about the issues of multiple reviews previously and especially explain about this in the bracket, which makes the justification less reasonable. Hope that the authors can better clarify this issue in the paper, avoid future misunderstanding, and release codes that are consistent with the statement and experimental results in the rebuttal.
> > > > > >
> > > > > > 2. There is an inconsistency here: the authors said "In this regard, none of the cited papers is comparable here", but they say that "nevertheless" they will provide the evaluation for the baselines. The point is quite vague here. Do the authors consider methods [1][2] possible baselines or not? I actually do not agree with their statement on not comparable, but at least the authors should make a clear point, so that I can give comments on whether they point is well supported by evidence. It is okay to admit that you were previously wrong about something, but please try to avoid misleading statements anymore just to avoid admitting that you were wrong.
> > > > > > Also, as I mentioned in the response, [1][2] are just two examples. Are there other retrieval-based methods? Please investigate more thouroughly, and I will help check whether there were missing one.
> > > > > >
> > > > > > I also to emphasize that the proposed method is not an end-to-end framework like [1][2], but relies on heavy heuristic post-processing for generating explanations.

---

> > > > > > > ### Author Response · Authors · 2022-11-24
> > > > > > > **Clarification on explanation generation details and evaluation**
> > > > > > >
> > > > > > > **Q1**: As shown by the statistics, it is very rare that a user writes multiple reviews about an item, so the original sentence is quite confusing. Hence I find it strange that the authors think about the issues of multiple reviews previously and especially explain this in the bracket, which makes the justification less reasonable. Hope that the authors can better clarify this issue in the paper, avoid future misunderstandings and release codes that are consistent with the statement and experimental results in the rebuttal.
> > > > > > >
> > > > > > > **R1**: We intended to inform the readers that there could be past reviews from (u,i) used as candidate reviews, even if the situation is rare. We didn’t realize that the additional explanation in brackets will instead cause misunderstanding. We will clarify this in the paper and thanks for letting us know about the issue. We will release our source code allowing others to replicate our results.
> > > > > > >
> > > > > > > **Q2**: There is an inconsistency here: the authors said "In this regard, none of the cited papers is comparable here", but they say that "nevertheless" they will provide the evaluation for the baselines. The point is quite vague here. Do the authors consider methods [1][2] possible baselines or not? Also, as I mentioned in the response, [1][2] are just two examples. Are there other retrieval-based methods? Please investigate more thoroughly, and I will help check whether there were missing ones.
> > > > > > > I also emphasize that the proposed method is not an end-to-end framework like [1][2], but relies on heavy heuristic post-processing for generating explanations.
> > > > > > >
> > > > > > > **R2**: Thanks for your provided references and efforts to ensure our paper is more rigorous in experimental evaluation. We did overlook the retrieval-based models which can be trained without relying on annotations and we will correct our inaccurate statements in the related work. As for the two papers [1][2], both of them proposed to generate explanations from item reviews. To make the generated explanations in the same form as ours for fair comparison in human evaluation, we additionally extracted sentences from user reviews with the highest attention weights as the explanations, even though there is no guarantee that the explanations from the item reviews and from the user reviews are about the same item. We generate explanations for the same samples used in our previous human evaluation and show the results below (the last two rows are copied from the Table A7 in revised paper for better comparison):
> > > > > > >
> > > > > > > | Model      |Relevance | Faithfulness  |  Informativeness |
> > > > > > > | :---        |    :----:   |          ---: |    ---: |
> > > > > > > | NARRE      |   2.67 |2.86 |3.86 |
> > > > > > > | WassersteinVAE   |   3.54 |2.84 |3.84 |
> > > > > > > | GIANT |  3.77 | 3.27| 4.17|
> > > > > > >
> > > > > > > The results show that NARRE can retrieve meaningful reviews (high informativeness), but achieve inferior results on selecting user-item pair reviews with overlapping aspects (low relevance).

---

> > > > > > > > ### Comment · Reviewer_Scqk · 2022-11-27
> > > > > > > > **Thanks & About points that are not fully addressed**
> > > > > > > >
> > > > > > > > Thank the authors for the explanations.
> > > > > > > > I did not increase the score currently because there are two points that have not been well addressed:
> > > > > > > > 1. Summary of retrieval-based related works beyond [1][2].
> > > > > > > > 2. A clearer introduction about the equations. The current version does not fully address my concerns stated in my original review.
> > > > > > > > If these two points are well addressed, I will consider increase my score.

---

> > > > > > > > > ### Author Response · Authors · 2022-12-03
> > > > > > > > > **Summary of existing retrieval-based explainable recommender**
> > > > > > > > >
> > > > > > > > > Retrieval-based (aka. extractive) solutions directly select representative text spans (rationales), i.e., words or sentences, from past reviews to explain the recommender behaviours and we focus on the recommenders with the ratings as the only supervision signal. As the user could rate/interact with diverse items, resulting in heterogeneous topics associated with different items, [3] proposed an asymmetrical hierarchical attention network to select sentences from user reviews that are highly similar to the target aspects. Concretely, it first calculates the attention for sentences in the target item reviews and then selects the sentences from a user review, which are highly similar to the most attentive sentences from item reviews. Many methods select rationales only from the target item’s existing reviews and attention-based methods are prevalent in review-based explainable recommenders [2,4,5,6,7,8,9,10,11,12,13]. Some of them argue the static and independent modeling for the user and item would fail to capture the relevance between the user’s preference and the item’s features: [8] proposed a dual attention mechanism to build dependency between the user and item; [11] introduced a time gate in GRU to control the change of user’s preference. [7,12] designed a hierarchical attention mechanism to capture multiple-grained information between the user and the item. [10] used a capsule network to select attentive sentences and [4] used a graph attention network, whose nodes include user, item,  review sentences, and aspect (aka. attribute) mentioned in their reviews, to select representative sentences by considering the user-item interaction and aspects overlappings.
> > > > > > > > >
> > > > > > > > > Beyond leveraging attention weights to select rationales supervised by the rating prediction results, [1,14,16,17] proposed different optimization objectives to refine the selection. [1] adopted reinforcement learning to retrieve the most relevant review sentences and impose *adjacency* and *sentiment-consistency* constraints on the selected sentences. [14] applied the similar adjacency constraint but it proposed a *representativeness* objective to select sentences that can represent other sentences with lower cost. [16] proposed to refine the extracted sentences in *length*, *content overlapping*, and *content rarity* via a decoder.  [17] argued that a promising rationale extract should be built on a minimal feature set but with maximum prediction ability. Accordingly, it proposed the L1 penalty to achieve the sparse feature set and minimize the prediction discrepancy derived from the selected rationale and all the review sentences, i.e., correlations. [15] leveraged K-means to cluster the user/item review sentences and used the assigned cluster centroids to derive the representation-explanation embedding, thus the sentences with the shortest distance to the centroid are extracted as rationales. None of the aforementioned approaches consider the user-item interaction information for building the explainable recommendation systems.
> > > > > > > > >
> > > > > > > > >
> > > > > > > > > References
> > > > > > > > > [1] A reinforcement learning framework for the explainable recommendation. 2018.
> > > > > > > > > [2] Neural attentional rating regression with review-level explanations. 2019.
> > > > > > > > > [3] Asymmetrical Hierarchical Networks with Attentive Interactions for Interpretable Review-Based Recommendation. 2020.
> > > > > > > > > [4] Graph-based Extractive Explainer for Recommendations. 2022.
> > > > > > > > > [5] Interpretable convolutional neural networks with dual local and global attention for review rating prediction. 2017.
> > > > > > > > > [6] A Context-Aware User-Item Representation Learning for Item Recommendation. 2017.
> > > > > > > > > [7] Hierarchical User and Item Representation with Three-Tier Attention for Recommendation. 2019.
> > > > > > > > > [8] Daml: Dual attention mutual learning between ratings and reviews for item recommendation. 2019.
> > > > > > > > > [9] Multi-Pointer Co-Attention Networks for Recommendation. 2019.
> > > > > > > > > [10] A Capsule Network for Recommendation and Explaining What You Like and Dislike. 2019.
> > > > > > > > > [11] Dynamic explainable recommendation based on neural attentive models. 2019.
> > > > > > > > > [12] Toward Comprehensive User and Item Representations via Three-tier Attention Network. 2022.
> > > > > > > > > [13] AENAR: An aspect-aware explainable neural attentional recommender model for rating prediction. 2022.
> > > > > > > > > [14] Synthesizing aspect-driven recommendation explanations from reviews(SEER). 2020.
> > > > > > > > > [15] Unsupervised Extractive Summarization-Based Representations for Accurate and Explainable Collaborative Filtering. 2021.
> > > > > > > > > [16] Comparative Explanations of Recommendations. 2022.
> > > > > > > > > [17] Accurate and Explainable Recommendation via Review Rationalization. 2022.

---

> > > > > > > > > > ### Comment · Reviewer_Scqk · 2022-12-05
> > > > > > > > > > **Keep my score**
> > > > > > > > > >
> > > > > > > > > > Thanks a lot to the authors for the much more comprehensive summarization of the related work. However, "None of the aforementioned approaches consider the user-item interaction information for building the explainable recommendation systems" is not really convicing to me. Thus, I will keep my score.

---

> > > > > > > > > ### Author Response · Authors · 2022-12-03
> > > > > > > > > **Reply to the first question about the equation in the original review**
> > > > > > > > >
> > > > > > > > > As for the two questions raised in the original review, we have replied in "**Response to Reviewer Scqk: About Information Bottleneck Theory**". We are not sure which part is not clear to you, so we give a more detailed deviation to the two questions here:
> > > > > > > > >
> > > > > > > > > **Q1**:
> > > > > > > > > (a) Why the second term in Eq. (3) is $I(X_t, Z_g)$ rather than $I(X_t, Z_t)$?
> > > > > > > > > (b) Why the first term considers only $Z_t$ but not $Z_g$?
> > > > > > > > > (c) Do you use Eq. (2) to derive Eq. (3)? Can you give a step-by-step derivation here? For example, can you first replace X, Z in Eq. (1) with $X_t$, $X_g$ and $Z_t$, $Z_g$ and gradually derive each equation?
> > > > > > > > >
> > > > > > > > > **R1**:  We firstly introduce the objective function of  Information Bottleneck as Eq. (1) as follows: (it was firstly proposed in paper [1], Eq. (3). )
> > > > > > > > > $$O_{IB} = I(\bar{X}, Z) - \beta I(Z, X)$$
> > > > > > > > > It is the Lagrange multiplier format for the object of maximising the mutual information $I(\bar{X},Z)$ subject to a constraint of $I(X,Z) < I_c$ (this was also elaborated in paper [1], Equation (2) ).
> > > > > > > > >
> > > > > > > > > If we consider only the modality of text, then the above equation can be written as:
> > > > > > > > > $$O_{IB} = I(\bar{X}^t, Z^t) - \beta I(Z^t, X^t)$$.
> > > > > > > > > Which aims to maximise the mutual information $I(\bar{X}^t,Z^t)$ subject to a constraint of $I(X^t,Z^t) < I_c$. **Our theory in Section 3 is to explain why we can maximise the same objective $I(\bar{X}^t,Z^t)$ by using a new constraint introduced from another modality**.
> > > > > > > > >
> > > > > > > > > According to Eq. (2), if the constraint of $I(X^t,Z^t) < l_c$ in Eq.(1) is true, there must be the constraint of $I(Z^g, X^t) < H + l_c$ which also exists. It means we can use information from the modality graph to satisfy the constraint. (The proof of Eq. (2) is in Appendix B in our revised paper and we will incorporate more explanations, e.g., how to apply chain rules, conditional entropy, in the next response **"Response to the second question about the derivation in section B"**)
> > > > > > > > >
> > > > > > > > > Considering the training of $I(X^t,Z^t)$ requires the prior of $Z^t$, which is not easy to get or set to be an over-simplified standard Gaussian, we choose the alternative constraint of $I(Z^g, X^t) < H + l_c$ which can be learned from the user-item interaction graph. Therefore, the learning objective becomes maximising the mutual information $I(\bar{X}^t,Z^t)$ subject to a constraint of $I(Z^g, X^t) < H + l_c$.
> > > > > > > > >
> > > > > > > > > We transfer it into Lagrange multiplier format, we can get the Eq. (3)
> > > > > > > > >  $$O_{IB} = I(\bar{X^t}, Z^t) - \beta I(Z^g, X^t)$$
> > > > > > > > >
> > > > > > > > > Here, I would like to clarify two points that could be confusing:
> > > > > > > > >
> > > > > > > > > (1) When we use the cross modality constraint, we have an additional term of $H$. However, $H$ is based on the input features which are irrelevant to learning. That’s why we can ignore $H$ in optimisation.
> > > > > > > > >
> > > > > > > > > (2) The constraints of $I(X^t, Z^t)$ and $I(X^t,Z^g)$ are different, i.e., $I_c$ and $H + I_c$), but one can be replaced with the other during optimisation. This is because in the Lagrange Multiplier method, the optimisation objective of the constraint term is based on the gradient of the constraint function. So, no matter what the constraint is, the gradient of $I_c$ or $H + I_c$ is equal to 0  in our final equation. That’s why we have a similar form of objective function even if we replace the constraint with another in the learning objective.
> > > > > > > > >
> > > > > > > > > That's how we derive the Eq.(3)
> > > > > > > > >
> > > > > > > > > ____
> > > > > > > > > Let’s go back to the question (a) and (b), why we keep the first term of $I(X_t, Z_t)$ but replace the $Z^t$ by $Z^g$ in the second term.
> > > > > > > > >
> > > > > > > > > According to the proof from the paper[1]:
> > > > > > > > > In Eq. (1), the first term of $I(\bar{X^t},Z^t)$ can be approximated by the reconstruction between $x_n^t$ and $\bar{x}_n^t$, which is easy to train. That’s why we keep this term in our final objective function Eq. (3).
> > > > > > > > > However, the second term of $I(X^t,Z^t)$ is approximated by $z_n^t$ and $r(Z^t)$, where $z_n^t$ is the learned hidden distribution of the input $x_n^t$ and $r(Z^t)$ is the prior of $z_n^t$, which is usually set to be a standard Gaussian distribution. Here, the difference between two distributions $z_n^t$ and $r(Z^t)$ is measured by KL divergence, which usually collapses if we do not set an appropriate prior distribution[2][3]. That’s why we want to use the posterior of $Z^g$ in our method instead of the prior of $Z^t$ to achieve a stable training process. Because the posterior of $Z^g$ is learnable from the graph.
> > > > > > > > >
> > > > > > > > > ----
> > > > > > > > > **References**
> > > > > > > > >
> > > > > > > > > [1] Alexander A. Alemi, Ian Fischer, Joshua V. Dillon, and Kevin Murphy. Deep variational information bottleneck. ICLR 2017.
> > > > > > > > >
> > > > > > > > > [2] VAE with a VampPrior. AISTATS 2018.
> > > > > > > > >
> > > > > > > > > [3] Adversarially Regularized Autoencoders. ICML2018.

---

> > > > > > > > > ### Author Response · Authors · 2022-12-03
> > > > > > > > > **Response to the second question about the derivation in section B**
> > > > > > > > >
> > > > > > > > > Q2: The proof in Section B is also difficult to understand. I am an expert in explainable recommendation but I am not an expert in information bottleneck. Every derivation step needs to be clearly written for me to carefully check the correctness of the proof, e.g., what is the chain rule, how you apply it twice.
> > > > > > > > >
> > > > > > > > > Response: First, we would like to explain the chain rule for the conditional entropy and how to apply it twice in our proof.
> > > > > > > > >
> > > > > > > > > In general, we assume we have three different variables $e_1$, $e_2$, and $e_3$. The joint entropy can be defined as: $H(e_1, e_2, e_3)$.
> > > > > > > > >
> > > > > > > > > The chain rule yields that $H(e_1, e_2, e_3) = H(e_1| e_2, e_3) + H(e_2|e_3) + H(e_3)$ is true.
> > > > > > > > >
> > > > > > > > > Here, the joint entropy is order irrelevant, which means that $H(e_1, e_2, e_3) = H(e_2, e_1, e_3)$. (We simply swap the positions of $e_1$ and $e_2$.)
> > > > > > > > >
> > > > > > > > > Then, we can apply the chain rule again for the joint entropy of  $H(e_2, e_1, e_3)$. As a result, $H(e_2, e_1, e_3) = H(e_2| e_1, e_3) + H(e_1|e_3) + H(e_3)$.
> > > > > > > > >
> > > > > > > > > Since $H(e_1, e_2, e_3) = H(e_2, e_1, e_3)$, we then have $H(e_1| e_2, e_3) + H(e_2|e_3) + H(e_3) =  H(e_2| e_1, e_3) + H(e_1|e_3) + H(e_3)$.
> > > > > > > > >
> > > > > > > > > After removing the $H(e_3)$ from both side, we have $H(e_1| e_2, e_3) + H(e_2|e_3) = H(e_2| e_1, e_3) + H(e_1|e_3)$
> > > > > > > > >
> > > > > > > > > Let’s take $e_1 = X^t$, $e_2 = X^g$, $e_3 = Z^g$, then we can obtain $H(X^t|X^g,Z^g) + H(X^g|Z^g) = H(X^g|X^t,Z^g) + H(X^t|Z^g)$
> > > > > > > > >
> > > > > > > > > Then, we move the $H(X^g|X^t,Z^g)$  to the left side of the above equation:
> > > > > > > > >
> > > > > > > > > $$H(X^g|Z^g) + H(X^t|X^g,Z^g) - H(X^g|X^t,Z^g) = H(X^t|Z^g)$$
> > > > > > > > >
> > > > > > > > > Then, by applying the following two inequalities to above equation:
> > > > > > > > >
> > > > > > > > > (1) Since the entropy must be positive, we have $H(X^t|X^g,Z^g) \geq 0$.
> > > > > > > > >
> > > > > > > > > (2) Since $Z^g$ and $X^g$ are dependent, according to the properties of conditional entropy, we have: $H(X^g|X^t,Z^g) \leq H(X^g|X^t)$.
> > > > > > > > >
> > > > > > > > > We then have $H(X^t|Z^g) \geq H(X^g|Z^g) - H(X^g|X^t)$
> > > > > > > > >
> > > > > > > > > By applying the properties of conditional differential entropy, which yields $H(X^g|Z^g) = H(X^g) - I(X^g,Z^g)$, the above formula can be simplified as:
> > > > > > > > >
> > > > > > > > > $H(X^t|Z^g) \geq H(X^g) - I(X^g,Z^g) - H(X^g|X^t)$
> > > > > > > > >
> > > > > > > > > Then, we can apply the above equation to the mutual information between $X^t$ and $Z^g$, where $I(X^t,Z^g) = H(X^t) - H(X^t|Z^g)$. Accordingly, we have
> > > > > > > > >
> > > > > > > > > $I(X^t,Z^g) = H(X^t) - H(X^t|Z^g) \leq H(X^t) - H(X^g) + I(X^g,Z^g) + H(X^g|X^t)$
> > > > > > > > >
> > > > > > > > > In our assumption, we assume that the $I_c$ is the constraint for the mutual information on both modalities of text and graph. (We can define the larger constraint on the two modalities as the uniform up boundary obviously)
> > > > > > > > >
> > > > > > > > > Then we have $I(X^t,Z^g) \leq H(X^t) - H(X^g) + H(X^g|X^t) + I_c$. If we take $H = H(X^t) - H(X^g) + H(X^g|X^t)$, which stands for the different between two modalities, we then have   $I(X^t,Z^g) \leq H + I_c$

---

> ### Author Response · Authors · 2022-11-18
> **Response to Reviewer Scqk: About Information Bottleneck Theory**
>
> **About Information Bottleneck Theory**:
>
> Q1: Why the second term in Eq. (3) is $I(X^t, Z^g)$ rather than $I(X^t, Z^t)$? Why the first term considers only $Z^t$ but not $Z^g$? Do you use Eq. (2) to derive Eq. (3)? Can you give a step-by-step derivation here? For example, can you first replace $X$, $Z$ in Eq. (1) with $X^t$, $X^g$ and $Z^t$, $Z^g$ and gradually derive each equation?
>
> R1: We updated Eq.1 and the descriptions around the Eq.1, Eq.2 Eq.3.  A step-by-step derivation is given in Appendix B. In the original information bottleneck (IB) theory, both the input $X$ and compressed latent variable $Z$ are from the same modality. For example, the variational framework is only associated with one modality data, text, then the input, compressed variable and reconstruct term can be denoted as $X^t, Z^t, \bar{X}^t$. However, the choice of a prior distribution over $Z$ is essential to learn robust representations.  In this paper, we prove that we can use the posterior distribution learnt from other modality, i.e., graph, as the prior, $Z^g$, but still maintain a new upper bound of the mutual information between input and latent variable. The derivation result, i.e., Eq. 3, is shown as follow, the right part is our derived new upper bound.
>
> $ I(X^t;Z^g) \leq H(X^t) - H(X^g) + H(X^g|X^t) + I_c $
>
> That’s why we replace the original $I(X^t;Z^t)$ by a cross domain term of $I(X^t;Z^g)$.  Eq.3 is derived from Eq.1 by using the results from Eq.2. Moreover, the evaluation part focuses on comparison with other variational models with different priors, i.e., Standard Gaussian Distribution and Wasserstein VAE,  to verify the efficiency of our proposed geometric prior.
>
> Q2: The proof in Section B is also difficult to understand.
>
> R2: Detailed derivation is shown in updated Appendix B.

---

### Official Review · Reviewer_JQQm · 2022-10-24

**Confidence:** 5
**Clarity, Quality, Novelty And Reproducibility:** Please refer to the strength and weak…
**Correctness:** 3
**Technical Novelty And Significance:** 2
**Empirical Novelty And Significance:** 2
**Recommendation:** 5

**Strength And Weaknesses:**

Strengths.
- The paper proposes an interesting idea to improve the interpretability of user/item latent factors in a recommender system by leveraging knowledge from both textual reviews as well as user/item clusters derived from user-item interaction signals.
- The proposed method offers competitive performance in terms of rating prediction as measured by MAE and RMSE metrics. Some qualitative/quantitative results also show that the proposed method seems to learn more coherent and meaningful latent representations and clusters.

Weaknesses.
- Writing requires some important improvements. Section 3 and 4 describing the proposed method are hard to follow.
(I) The notations are a bit confusing. I would recommend using “t” and “g” to denote text and graph related quantities as superscript instead of subscript.
(II) The transition from eq. 2 to eq. 3 is not obvious. An alternative is to introduce eq. 4 first, which corresponds to a Beta-VAE with a cluster-membership distribution as a prior, and then provide a connection to the IB objective as a supportive analysis for eq. 4.
(III) Eq. 8 regarding the KL-term is not obvious either, is p(Z_t|x_n,e) Gaussian?
- The focus of this work is on explainability, however the results regarding this aspect are weak. I would recommend including some human evaluations (user study) to better assess the quality of the explanations generated by proposed method compared to the baselines.
- The performance of the proposed recommender system is measured using prediction metrics such MAE and RMSE. In general, MAE and RMSE do not necessarily reflect the quality of item recommendation. Reporting retrieval measures such as Precision and Recall would be more convincing.

Additional comments.
- The clustering component is central in the proposed method. It would be useful to have more experiments regarding this aspect. For instance, what is the impact of number of clusters on the performance of the proposed method.


**Summary Of The Paper:**

This paper proposes an explainable recommender system based on variational autoencoders (VAE). The main idea is to model user/item textual reviews with an autoencoder whose latent space is regularized by user/item cluster-membership distributions. The latter are derived from the user-item interaction data by first clustering users and items, and then estimating a user or item distribution based on its distance to each cluster centroid using a Gaussian Kernel. The authors argue that such regularization would enable the use of collaborative signals from similar users and items while generating explanations for a given user-item pair. Experiments are curried out on three real-world datasets. Various aspects of the proposed method are evaluated, including rating prediction and latent variable interpretability.

**Summary Of The Review:**

Please refer to the strength and weaknesses section above.

---

> ### Author Response · Authors · 2022-11-18
> **Response to Reviewer JQQm**
>
> Q1. Writing requires some important improvements. Section 3 and 4 describing the proposed method are hard to follow.
>
> R1. Notations have been updated with $t$, $g$ used as superscripts.  $p(Z^t|x^t_n,\epsilon) $ is not a Gaussian distribution, instead, it is the cluster assignment distribution learnt from a user-item graph. Our primary contribution is to leverage prior learnt from graph to replace the Gaussian prior for better *transferability*, *practicality* and *interpretability*. Please refer to updated Section 3 and Appendix B.
>
> Q2.  I would recommend including some human evaluations (user studies) to better assess the quality of the explanations generated by the proposed method compared to the baselines.
>
> R2. Human evaluation results are added to Appendix E.
>
> Q3. Reporting retrieval measures such as Precision and Recall would be more convincing. For instance, what is the impact of the number of clusters on the performance of the proposed method?
>
> R3. Precision and Recall results are added to Appendix D.1. Please refer to Appendix D.2. for the effects of different number of clusters.

---

> > ### Comment · Reviewer_JQQm · 2022-11-28
> > **Comment after author response**
> >
> > I thank the authors for response and efforts to improve the paper. The authors’ response addresses partially my original concerns regarding writing, human evaluation, and recommendation accuracy in terms of Precision and Recall. Some limitations remain:
> >
> > - The newly added human evaluation is an important improvement, yet it is weak given that explainability is the main point of the paper.  It is hard to draw conclusions based on one dataset only and on three evaluators. Given the topic of the paper, one would expect explainability evaluation to be the focus of the experiments.
> > - It would be more realistic to report Precision and Recall at recommendation lists with more than one element (e.g., consider recommendation lists of size 10 instead of 1 currently)
> > - The new notations improve the readability of section 3 and 4. The transition from eq. 2 to eq. 3 remains not obvious.
> > - Regarding the prior learned from the user-item graph isn’t it $r(Z^g)$? My question is about $p(Z^t|x_n^t, \epsilon)$, which seems to be another categorical distribution if my understanding is correct?

---

> > > ### Author Response · Authors · 2022-12-03
> > > **Response of "more human evaluation results" and "larger recommendation lists"**
> > >
> > > **Q1**: Human evaluation results on one dataset from three evaluators are weak.
> > >
> > > **R1**: As  *Beer* and *Music* require prior knowledge on the specific products, we spend some time finding appropriate evaluators with relevant interests in the two areas. The updated results are shown as below:
> > > | **Dataset** | **Model**      | **Relevance** | **Faithfulness** | **Informativeness** |
> > > |-------------|----------------|---------------|------------------|---------------------|
> > > | Beer        | NARRE          | 2.7           | 2.4              | 3.1                 |
> > > |             | WassersteinVAE | 3.6           | 3.5              | **3.5**                 |
> > > |             | GIANT          | 3.5           | **3.7**              | **3.5**                 |
> > > | Music       | NARRE          | 4.4           | 3.2              | 3.8                 |
> > > |             | WassersteinVAE | 4.0          | 3.1              | 3.0                   |
> > > |             | GIANT          | **4.7**           | **3.8**              | **4.1**                 |
> > >
> > > **Q2**: It would be more realistic to report Precision and Recall at recommendation lists with more than one element.
> > >
> > > **R2**: If we increase the size of the recommendation list, **GIANT** still outperforms other VAE-based models in both precision and recall on the Beer dataset. But its benefit over **NARRE** diminishes. However, on the *Music* and *Office* datasets, **GIANT** gives superior performance compared to others in general, with more obvious improvement on the *Office* dataset.
> > >
> > > | **Dataset** | **Metrics** | **NARRE** | **AutoEncoder** | **VAE** | **WassersteinVAE** | **GIANT** |
> > > |-------------|-------------|-----------|-----------------|---------|--------------------|-----------|
> > > | Beer        | Pre@3       | **0.58**      | 0.42            | 0.46    | 0.47               | 0.47      |
> > > |             | Recall@3    | **0.54**      | 0.41            | 0.46    | 0.48               | 0.49      |
> > > |             | Pre@5       | **0.64**      | 0.49            | 0.52    | 0.53               | 0.52      |
> > > |             | Recall@5    | 0.60       | 0.48            | 0.54    | 0.54               | **0.61**      |
> > > |             | Pre@10      | **0.64**      | 0.49            | 0.52    | 0.54               | 0.56      |
> > > |             | Recall@10   | 0.61      | 0.52            | 0.55    | 0.55               | **0.62**      |
> > > | Music       | Pre@3       | **0.50**       | 0.46            | 0.49    | 0.46               | **0.50**       |
> > > |             | Recall@3    | 0.51      | 0.47            | 0.53    | 0.51               | **0.54**      |
> > > |             | Pre@5       | 0.59      | 0.58            | **0.60**     | 0.55               | **0.60**       |
> > > |             | Recall@5    | 0.62      | 0.60             | **0.65**    | 0.62               | **0.65**      |
> > > |             | Pre@10      | 0.62      | 0.60            | 0.62    | 0.60                | **0.63**      |
> > > |             | Recall@10   | 0.65      | 0.62            | 0.69    | 0.68               | **0.69**      |
> > > | Office      | Pre@3       | 0.51      | 0.47            | **0.60**     | **0.60**                | 0.58      |
> > > |             | Recall@3    | 0.49      | 0.50             | 0.52    | 0.53               | **0.55**      |
> > > |             | Pre@5       | 0.59      | 0.61            | **0.67**    | **0.67**               | **0.67**      |
> > > |             | Recall@5    | 0.58      | 0.60             | 0.61    | 0.61               | **0.63**      |
> > > |             | Pre@10      | 0.60       | 0.62            | **0.68**    | 0.67               | **0.68**      |
> > > |             | Recall@10   | 0.59      | 0.61            | 0.61    | 0.61               | **0.64**      |

---

> > > ### Author Response · Authors · 2022-12-03
> > > **Response of "Transition from eq.2 to eq.3 remains not obvious"-simplified version**
> > >
> > >
> > > In Eq.(1), the learning objective of maximising the mutual information $I(\bar{X},Z)$ with a constraint of $I(X,Z) < I_c$ can be optimised by the Lagrange Multiplier method: $I(\bar{X},Z) - \beta I(X,Z)$.
> > >
> > > If we focus on text-based features, the above can be written as  I(\bar{X^t},Z^t) - \beta I(X^t,Z^t).
> > >
> > > However, the constraint term of $I(X^t,Z^t)$ is difficult to train. Therefore, in Eq.(2), we prove that the constraint of $I(X^t,Z^t) < l_c$ equals the constraint of $I(X^t,Z^g) < H + l_c$, proof can be found in Appendix B.
> > >
> > > Then, Eq.(1) can be approximated by the equivalent objective function of optimising the mutual information $I(\bar{X^t},Z^t)$ with a constraint of $I(X^t,Z^g) < H + I_c$, which can be written as  $I(\bar{X^t},Z^t) - \beta I(X^t,Z^g)$ by the Lagrange Multiplier method.
> > >
> > > Here, there are two points that might be confusing:
> > >
> > > When we use the cross modality constraint, we have an additional term of H. However, H is based on the input features which are irrelevant to learning. That’s why we can ignore H in optimisation.
> > >
> > > The constraints of $I(X^t, Z^t)$ and $I(X^t,Z^g)$ are different, i.e., $I_c$ and $H + I_c$, but one can be replaced with the other during optimisation. This is because in the Lagrange Multiplier method, the optimisation objective of the constraint term is based on the gradient of the constraint function. So, no matter what the constraint is, the gradient of $I_c$ or $H + I_c$ is equal to 0 in our final equation. That’s why we have a similar form of objective function even if we replace the constraint with another in the learning objective.

---

> > > ### Author Response · Authors · 2022-12-03
> > > **Response of "Transition from eq.2 to eq.3 remains not obvious"-Detailed version**
> > >
> > > -----------
> > > **1. Original Information Bottleneck in Single Modality**
> > >
> > > Eq. (1) is the learning objective of the information bottleneck theory, firstly proposed in paper [1], which yields that the ideal optimisation of an encoder-decoder structure $X \xrightarrow[]{encode} Z \xrightarrow[]{decode} \bar{X}$. **It can be approximated by maximising the mutual information $I(\bar{X},Z)$ with a constraint of $I(X,Z) < I_c$ ( from the Eq (2) in paper [1])**.
> > >
> > > Here, the main idea of this objective function is that, for any two examples, $x_1$ and $x_2$, from the training set:
> > >
> > > (1) On the one hand, to learn the encoding variable $Z$ which can maximally represent $\bar{x}$ in the reconstruction process (for example, we can use point-wise information to measure it) between the two examples, that is, $PMI(z_1,z_2) = PMI(\bar{x_1},\bar{x_2})$, which indicates a large mutual information of $I(\bar{X},Z)$ between this two distributions.
> > >
> > > (2) On the other hand, the encoding process should maximally compress the information of $x$, i.e.,  $PMI(z_1,z_2) = PMI(x_1,x_2) + \epsilon$, where $\epsilon$ is the Gaussian noise, which indicates a constraint on the mutual information $I(X,Z) \leq I_c$ between this two distributions.
> > >
> > > To optimise the above objective function, we use the Lagrange Multiplier method, which can be summarised as follows: in order to find the maximum of a function $f(x)$ subject to the constraint $g(x) < c$, the Lagrangian function can be written as: $f(x) - \beta g(x)$
> > >
> > > Therefore, the objective function in our method can be written as:
> > > $I(\bar{X}, Z) - \beta I(Z, X)$, where the first term is based on the reconstruction error and the second term is based on the prior of latent/encoding representation $Z$. For the  text modality, the above objective function can be written as $I(\bar{X^t}, Z^t) - \beta I(Z^t, X^t)$.
> > >
> > > --------
> > >
> > > **2. Our proposed Information Bottleneck by using a new constraint from graph modality**
> > >
> > > Now, let’s move to the next equation of Eq.(2). In Eq.(1), we found that the second term of $I(Z^t, X^t)$ is difficult to train if the prior $Z^t$ is not set properly ( the issue has attracted lots of attention, for example, in paper [2][3]). Hence, we assume that we have the posterior of $Z$ learned from another modality, in our case, Z^g from the user-item graph. For any two samples $x_1$ and $x_2$, there should be $PMI(x^t_1,x^t_2) \sim PMI(x^g_1,x^g_2)$, even if they come from different modalities. Because if two samples are similar in one modality, they should be similar in the other modalities as well.
> > >
> > > Considering the information bottleneck in these two modalities, we have $PMI(x^t_1,x^t_2) = PMI(z^t_1,z^t_2) + \epsilon^t$ and $PMI(x^g_1,x^g_2) = PMI(z^g_1,z^g_2) + \epsilon^g$. Therefore, it is reasonable to assume that there should be some constraint $h$ satisfying that $PMI(x^t_1,x^t_2) = PMI(z^g_1,z^g_2) + h$.  In Appendix B, we prove this constraint does exist: if the constraint of $I(X^t,Z^t) < l_c$ is true, there must be the constraint of $I(X^t,Z^g) < H + l_c$ which also exists (As requested by Reviewer Scqk, the step-by-step derivation, including how to apply chain rule and conditional entropy, etc, can be found in **Response to the second question about the derivation in section B** ).
> > >
> > > Then, let’s move to the original IB objective in Eq.1 in the single-modality (only text), which is defined as maximising the mutual information $I(\bar{X^t},Z^t)$ with a constraint of $I(X^t,Z^t) < I_c$. Then, we can replace the constraint by an equivalent one: maximising the mutual information $I(\bar{X^t},Z^t)$ with a constraint of $I(X^t,Z^g) < H + I_c$.
> > >
> > > Then, we can rewrite the above objective function by the Lagrange Multiplier method:
> > >
> > > $$I(\bar{X^t},Z^t) - \beta I(X^t,Z^g)$$
> > >
> > > Here, we can simply replace the $I(X^t,Z^t)$ by $I(X^t,Z^g)$ because the **Lagrange Multiplier method is based on the gradient of constraint $( I(X^t,Z^t) -  l_c < 0$ and $I(X^t,Z^g) - H - I_c < 0)$, where the gradient of $I_c$ and $H + I_c$ are both 0**, so we are able to focus on the non-zero term of  $I(X^t,Z^t)$ (in Eq.(1)) or $I(X^t,Z^g$ (in Eq.(3)) only.
> > >
> > >
> > > References
> > >
> > > [1] Alexander A. Alemi, Ian Fischer, Joshua V. Dillon, and Kevin Murphy. Deep variational information bottleneck. ICLR 2017.
> > >
> > > [2] VAE with a VampPrior. AISTATS 2018.
> > >
> > > [3] Adversarially Regularized Autoencoders. ICML2018.

---

> > > ### Author Response · Authors · 2022-12-03
> > > **Response to question about the "prior learned from graph"**
> > >
> > > **Q**: Regarding the prior learned from the user-item graph isn’t it $r(Z^g)$? My question is about $p(Z^t|x_n^t,\epsilon)$, which seems to be another categorical distribution if my understanding is correct?
> > >
> > > **R**: Yes, your understanding is totally correct. $r(Z^g)$ is the prior learnt from user-item graph, in many existing work, it is set to be standard Gaussian distribution. $p(Z^t|x_n^t,\epsilon)$ is the derived posterior distribution in our information bottleneck, i.e., the VAE framework with  reviews as input. We regard each dimension in the latent variable $z^t$ as a cluster, so the distribution $Z^t$ is a soft categorical distribution, whose value represents the possibility of this input sample being assigned to this cluster/dimension.
> > >
> > > To guarantee each dimension from $Z^g$ and $Z^t$ is related to the same category, we propose a method which is called Prior-centralisation. In this method, we use the posterior of $Z_g$ as a constraint to regularise both the initialisation and optimisation to guarantee that each dimension from these two modalities is aligned during the whole training process.

---

### Author Response · Authors · 2022-11-18
**General response**

The common concern from all the reviewers is the explanation generation and interpretability evaluation. Our responses are as follows:

In the submitted version, our generated interpretations are essentially the representative words in each user/item cluster, i.e., a dimension of the latent variable $z_{n}$. We have evaluated the interpretability by evaluating the generated clusters (i.e., the learned latent variables) from various aspects:
1. The inter- and intra- cluster metrics, i.e., the semantically coherence within a cluster and the semantic separation across clusters (See the Section: cluster separability and coherence (moved to Appendix D.4 in the revised version)).
2. The faithfulness of our generated explanations (aka. Completeness in the Section *completeness evaluation by perturbing on latent variables*). The results show that removing the representative words in the latent dimensions identified by our models leads to larger prediction results changes, which implies the importance of our identified words in model rating prediction.
3. A case study to show that our generated interpretations are intuitive and informative, as they can summarize the interests/features of users/items in different aspects, as well as their relative impacts on model prediction (See Figure 4 in the initial submission, clusters of a larger size are more important).

We agree that the extracted words can be noisy and not related to the specific user-item pair, and human evaluation results can better evaluate the quality of the generated explanations. Therefore, we extract sentences from user/item reviews that are most relevant to the user/item latent topics, i.e., the representative words as the interpretations shown to the evaluators (see the updated explanations in Figure 1 and Figure 4 in the revised version). The detailed  interpretation generation process and the human evaluation results in terms of *relevance*, *faithfulness* and and *informativeness* are shown in Appendix E.

---
All the updates are highlighted in blue in the revised paper, including the following changes to address the reviewers’ concerns:

1. Updated the description of related work at the end of Section 2: related work.
2. Updated the derivation of Eq.1 and Eq.3 about information theory in Section 3.
3. Updated the proof of Eq.2 in Appendix B.
4. Updated the rating prediction module description round Eq.11 and in Appendix A.2 Hyper-parameter Setting.
5. Added the interpretation generation process in Appendix E.
6. Updated the generated interpretations examples in Figure 1 and Figure 4 (as a case study).
7. Added the human evaluation process and results in Appendix E.
8. Added evaluation results on Precision@1 and Recall@1 in Appendix D.1.
9. Added the experimental results of different numbers of clusters in Appendix D.2.
10. Added the Wasserstein VAE results in Sections 6.1, 6.2, and 6.3.

---

### Decision · Program_Chairs · 2023-01-20

**Decision:**

Reject

**Justification For Why Not Higher Score:**

There have been a lot of discussions around human evaluation on the generated explanation. I unfortunately do not have the expertise around it to make a fair judgement. (Maybe a more domain-specific venue would also be a good option here.) However, from a technical point of view, I think the writing of the paper can be further improved to be more concise. Furthermore, some of the design choices can probably be justified a bit better. Finally, please move away from rating prediction task.

**Justification For Why Not Lower Score:**

N/A

**Metareview: Summary, Strengths And Weaknesses:**

Summary: This paper presents an explainable recommender systems with $\beta$-VAEs where the prior is from the user-item interaction graph while the approximating posterior and the data-likelihood model is on the review text data. In this way, the information can be transferred between different modalities. Specifically, a GIANT framework is build to facilitate the VAE formulation by clustering users and items and smoothing out the density through kernels. The proposed model is able to achieve good rating prediction performance as well as generating explainable recommendations.

Strengths: All the reviewers agree that the paper proposes an interesting idea to improve the interpretability of the latent space in a recommender system by leveraging data from multiple modalities. The rating prediction results are promising (though I have bigger concern about this below).

Weaknesses: Besides mixed opinions on the human evaluation, there doesn't appear to be a consensus on the weaknesses of the paper: Two of the reviewers found that the writing requires some improvement while the other reviewer thinks the paper is well written. I think this will highly depend on the reader's background -- I consider myself very familiar with the general recommender systems as well as the VAE/information bottleneck line of work, but not very much into explainability and I found the paper not an easy read. I think the high-level idea of the paper is quite straightforward (maybe even lacks a bit technical novelty) but some of the writing feels a bit contrived -- the overall $\beta$-VAE formulation makes sense, but it doesn't have to draw into information bottleneck. Furthermore, section 4 reads to me rather busy with many design choices seemingly arbitrary without adequate justification.

As I mentioned earlier, I am no expert on explainability so I will not judge on the human evaluation. However, I do have a lot to say about rating prediction: Training (and evaluating) with RMSE/MAE on the observed ratings assumes all the missing ratings are missing at random, which is clearly not true (see Marlin et al. 2007, Collaborative Filtering and the Missing at Random Assumption) and has been shown to lead to biased results even when you get good RMSE/MAE (see, e.g., Steck 2010, Training and testing of recommender systems on data missing not at random, Schnabel et al. 2016, Recommendations as Treatments: Debiasing Learning and Evaluation). Furthermore, a model with good RMSE in a lot of cases does not translate to good recommendations (Cremonesi et al. 2010, Performance of recommender algorithms on top-n recommendation tasks). Therefore, in general whenever I see a paper which focuses on rating prediction, I have to point this out because I think the field as a whole should move away from it. One possible direction is to take an implicit feedback approach (e.g., Liang et al. 2018, Variational autoencoders for collaborative filtering), or a more causal based approach (e.g., Schnabel et al. 2016, Recommendations as Treatments: Debiasing Learning and Evaluation; Wang et al., 2020, Causal inference for recommender systems).